# Ultrasensitive detection of clinical pathogens through a target-amplification-free collateral-cleavage-enhancing CRISPR-CasΦ tool

Huiyou Chen [1,2,3,12], Fengge Song [1,2,3,12], Buhua Wang [1,3], Hui Huang[4], Yanchi Luo[1,3], Xiaosheng Han[4], Hewen He[5], Shaolu Lin[5], Liudang Wan[5], Zhengliang Huang[5], Zhaoyong Fu[5], Rodrigo Ledesma-Amaro [6], Dapeng Yin[7], Haimei Mao[8], Linwen He[9], Tao Yang[10], Zijing Chen[10], Yubin Ma[10], Evelyn Y. Xue[10], Yi Wan [1,3,11] ✉ & Chuanbin Mao [10] ✉

Clinical pathogen diagnostics detect targets by qPCR (but with low sensitivity) or blood culturing (but time-consuming). Here we leverage a dual-stem-loop DNA amplifier to enhance non-specific collateral enzymatic cleavage of an oligonucleotide linker between a fluophore and its quencher by CRISPR-CasΦ, achieving ultrasensitive target detection. Specifically, the target pathogens are lysed to release DNA, which binds its complementary gRNA in CRISPR-CasΦ to activate the collateral DNA-cleavage capability of CasΦ, enabling CasΦ to cleave the stem-loops in the amplifier. The cleavage product binds its complementary gRNA in another CRISPR-CasΦ to activate more CasΦ. The activated CasΦ collaterally cleaves the linker, releasing the fluophore to recover its fluorescent signal. The cycle of stem-loop-cleavage/CasΦ-activation/fluorescence-recovery amplifies the detection signal. Our target amplification-free collateral-cleavage-enhancing CRISPR-CasΦ method (TCC), with a detection limit of 0.11 copies/μL, demonstrates enhanced sensitivity compared to qPCR. It can detect pathogenic bacteria as low as 1.2 CFU/mL in serum within 40 min.

An estimated number of 13.7 million deaths globally in 2019 was associated with pathogenic infections, with 7.7 million of these linked to 33 common pathogens[1,2]. Five major bacterial pathogens, *Staphylococcus aureus*, *Escherichia coli*, *Streptococcus pneumoniae*, *Klebsiella pneumoniae*, and *Pseudomonas aeruginosa*, accounted for 54.9% of the surveyed bacterial infection-related deaths. Among them, bloodstream infections (BSI)[3] are common with over tens of millions of cases per year. Rapid, efficient, and sensitive molecular diagnostics are critical for early prevention and treatment of these infections[4,5]. Quantitative polymerase chain reaction (qPCR)[6] and culture methods[7] remain

[1]State key laboratory of digital medical engineering, Hainan University, Haikou 570228, China. [2]School of Life and Health Sciences, Hainan University, Haikou 570228, China. [3]State Key Laboratory of Marine Resource Utilization in South China Sea, Hainan University, Haikou 570228, China. [4]Microbial Medical Laboratory, People's Hospital of Haikou, Haikou 570208, China. [5]Hainan Viewkr Biotechnology Co., Ltd, Haikou 570228, China. [6]Imperial College Centre for Synthetic Biology, Department of Bioengineering, Imperial College London, London, United Kingdom. [7]Hainan Center for Disease Control and Prevention, Haikou 570228, China. [8]Products Quality Supervision and Testing Institute of Hainan Province, Haikou 570003, China. [9]School of Marine Biology and Fisheries, Hainan University, Haikou 570228, China. [10]Department of Biomedical Engineering, The Chinese University of Hong Kong, Shatin, Hong Kong SAR, China. [11]School of Biomedical Engineering, Hainan University, Haikou 570228, China. [12]These authors contributed equally: Huiyou Chen, Fengge Song. ✉e-mail: 993602@hainanu.edu.cn; cmao@cuhk.edu.hk

the gold standards for clinical bacterial pathogen diagnosis in blood or other samples. However, for ultra-low pathogen levels in clinical samples (e.g., blood may contain only 1 to 2 CFU/ml)[8–12], the low amount of DNA available after lysis of bacteria makes it difficult for qPCR (Sensitivity is usually $0.1 \times 10^4 – 10^5$ copies/mL)[13–15] to detect the pathogens through the quantification of their genome. Numerous studies report that the molecular diagnostic methods based on PCR technology to detect BSI pathogen have sensitivity of 43–99% and specificity of 60–100%[16–20]. The blood culture method relies on a lengthy culturing process (up to one week) to reach an increased and detectable pathogen level. For the time-consuming blood culture method, the growing bacteria need to be detected by either antimicrobial susceptibility testing of bacterial suspension or Matrix Assisted Laser Desorption Ionization-Time Of Flight Mass Spectrometer (MALDI-TOF MS) analysis of bacterial clonies formed after additional one-day plate culturing[16]. Among current diagnostic methods, T2 Magnetic Resonance Technology (T2MR) combined with PCR stands out as the only FDA-approved commercial solution for whole blood sample diagnosis[20]. This approach has a detection time ranging from 3 to 7 h, significantly reducing the diagnostic timeframe for BSI in clinical whole blood samples[21,22]. However, T2MR technology requires specialized reagent kits and related equipment. Moreover, it relies on asymmetric PCR pre-amplification[20], thus its applicability is not widespread. As noted, qPCR suffers from high false-negative rates due to limited sensitivity[3,23], while culture methods combined with MALDI-TOF MS require at least 3 days and exhibit relatively low positive rates[7], representing bottlenecks in current clinical pathogen diagnostics. Consequently, the challenge lies in developing a high-sensitivity, rapid, and efficient integrated point-of-care (POC) diagnostic approach for pathogen detection in clinical settings.

There is presently a lack of POC diagnostic tools enabling rapid, precise, and efficient analysis of pathogenic bacteria in clinical applications[24]. An ideal POC diagnostic should concurrently exhibit: (1) aM level gene detection or single pathogenic cell analysis without pre-amplification; (2) rapid signal output from sample to result within 1 h, and (3) one-pot, accurate clinical analysis at low cost. Based on clustered regularly interspaced short palindromic repeats (CRISPR), CRISPR-associated protein (Cas) systems have opened up new avenues for molecular diagnostics of nucleic acids[25]. Among them, CRISPR diagnostics strategies represented by DETECTR and SHERLOCK provide a mainstream POC design idea[26,27], and can be developed into portable, convenient, and efficient diagnostic tools to serve resource-limited areas[28–30]. However, these CRISPR-associated technologies rely on isothermal amplification methods such as loop-mediated isothermal amplification (LAMP) and recombinase polymerase amplification (RPA) to pre-amplify nucleic acid molecules[31,32]. SHINE[28,33] and STOPCovid[13] represent significant advances in one-pot reactions of pre-amplification combined with CRISPR systems. However, their reaction systems are complex, and primer design could be more convenient. Therefore, To overcome the above limitations, the mainstream strategy is to improve the sensitivity without pre-amplification, aiming to eliminate reaction risks affecting CRISPR system detection[34].

Under this direction of improvement, there are 3 strategies to enhance the detection performance of CRISPR-Cas: (1) Engineering Cas proteins and guide RNAs (gRNAs) based on the high programmability and optimization potential of CRISPR-Cas systems. For example, mutating the enzymatic activity sites of Cas12 and Cas13 proteins to increase their non-specific collateral-cleavage (i.e., *trans*-cleavage) activity[14,35,36], while the gRNAs can also be engineered and screened[37,38] to improve detection sensitivity or specificity. (2) Improving the detection sensitivity by enriching or concentrating the analyte. For example, combining microfluidics and microdroplet platforms to concentrate and enrich the reaction system can achieve high throughput, miniaturization, portability, and improved sensitivity[30,39,40]. In addition, nucleic acid analytes can be pre-concentrated or enriched, nucleic acid extraction steps optimized to increase the absolute copy number or concentration of targets[41–44]. (3) Optimizing and designing CRISPR systems through enzyme cascading amplification strategies to achieve one-pot, amplification-free detection performance. In recent years, researchers have designed and developed strategies for cascading reactions of Cas proteins with other enzymes[29,41,42,45,46]. So far, among the reported CRISPR-based gene detection, the detection limit ranges from thousands of copies to 0.5 copies per µL. More importantly, most strategies can achieve one-pot, amplification-free, ultra-sensitive detection performance, but they still cannot simultaneously meet the requirements of high specificity, high sensitivity, rapidity, convenience, and low cost in clinical applications, especially BSI.

To fulfill the aforementioned criteria for one-pot, rapid POC diagnosis, designing amplification-free and collateral-cleavage-enhancing (CCE) reactions may be the most suited approach. Cas proteins (Cas12 and Cas13 family) can exhibit both specific and non-specific enzymatic cleavage capabilities, and such non-specific cleavage (collateral *trans*-cleavage) capability can only be activated when gRNA binds the complementary DNA or RNA. Inspired from this unique feature of CasΦ (a member of Cas12 family), we propose a CCE method to enhance the non-specific collateral CasΦ cleavage of the oligonucleitide linker (between the fluorescent quencher and reporter) by designing and optimizing a DNA (with dual stem-loop blocking domains) that serves as an amplifier (Fig. 1). CasΦ (also known as Cas12j) is a type V protein with an approximate molecular weight of 80 kDa[47]. It harbors a RuvC-like structural domain and *trans*-cleavage activity akin to the Cas12 subfamily yet exhibits less than 7% amino acid sequence identity to other type V proteins, being solely associated with the TnpB nuclease superfamily[36,48]. After lysis of the target pathogens to release their DNA, the target DNA binds the complex of gRNA1 and CasΦ (ribonucleoprotein complex 1, RNP1) to activate the collateral cleavage activity of CasΦ, leading to the cleavage of the dual stem-loops in the amplifier. The resultant cleavage products bind gRNA2 (recognizing the products), activating the collateral cleavage activity of CasΦ in ribonucleoprotein complex 2 (RNP2). The activated CasΦ cleaves a large amount of the oligonucleotide linkers between the reporter and quencher, releasing the reporter to recover fluorescent signal. The cycle of stem-loop-cleavage/CasΦ-activation/fluorescence-recovery amplifies the fluorescent detection signal even when the target pathogen has an ultra-low level in the clinical samples such as serum. Our method, termed TCC (Target-amplification-free Collateral-cleavage-enhancing CRISPR-CasΦ method) reaches a record-low detection limit of 0.18 aM and took only 40 min. In terms of detecting the genes of target pathogens, its sensitivity is superior to qPCR. It exhibits ultra-sensitivity, allowing for the detection of pathogen loads (*S. aures, P. aeruginosa, K. pneumoniae, E. coli*) as low as 1.2 CFU/mL in clinical samples collected from BSI patients.

## Results
### Working principle of one-pot, collateral-cleavage-enhancing TCC
TCC achieves one-pot isothermal amplification-free POC diagnosis through enzymatic CCE amplification enabled by a tailored DNA signal amplifier for CasΦ, termed the TCC amplifier (Fig. 1). The reaction components are minimal, comprising CasΦ, gRNA, the TCC amplifier, a fluorescent reporter linked with a quencher, and microbial thermal lysate containing the target genome, enabling streamlined one-pot diagnosis (Fig. 1a). RNP1 is assembled from the guide RNA (gRNA1) and CasΦ, recognizing and targeting the genomic DNA (target). RNP2 is assembled from the gRNA2 and CasΦ, recognizing and targeting the cleaved product of the TCC amplifier. The unmodified single-stranded DNA (ssDNA) TCC amplifier folds into double-stranded DNA (dsDNA) with two stem-loop structures upon annealing, residing in a thermo-dynamically state before loop cleavage. Upon cleavage of both loops,

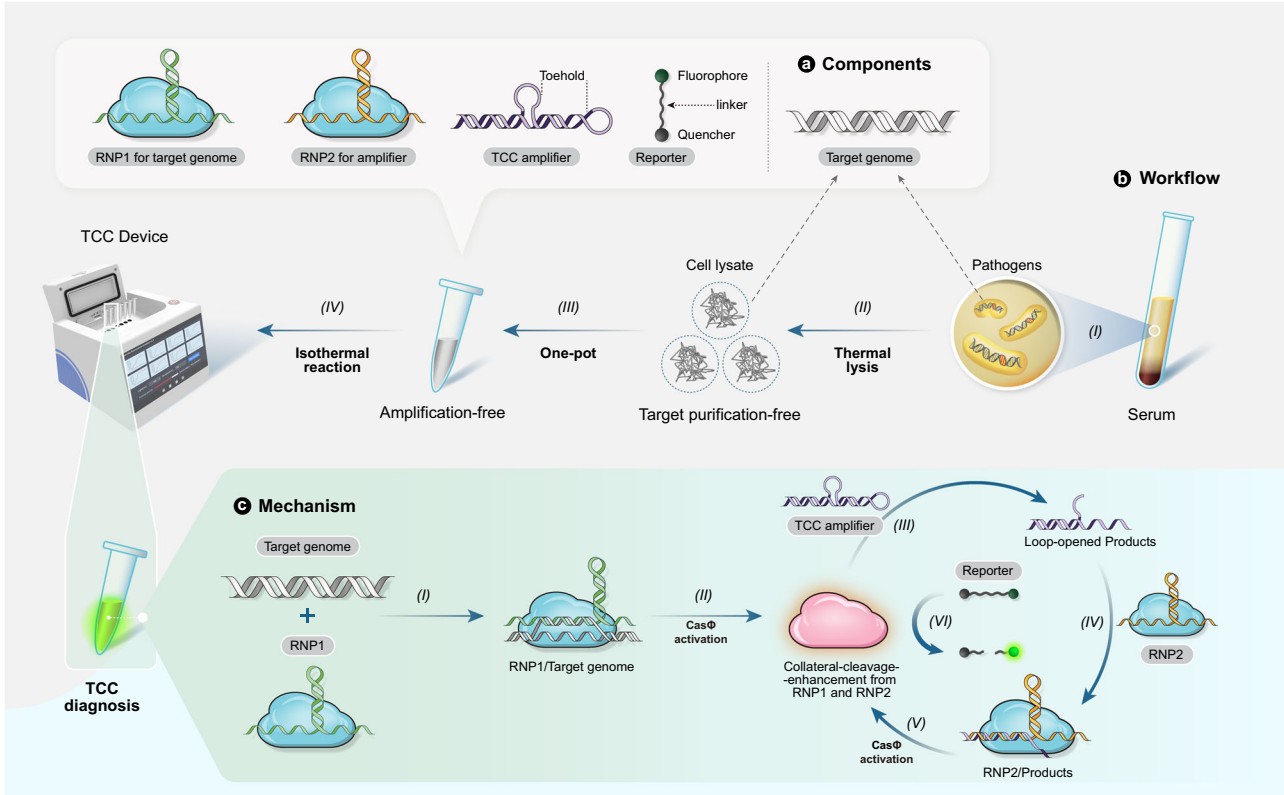

**Fig. 1 | Illustration of TCC (Target-Amplification-free Collateral-cleavage-enhancing CRISPR-CasΦ method) workflow and operating principles. a** TCC components. The one-step TCC reaction mixture comprises ribonucleoprotein complexes RNP1 and RNP2, an TCC amplifier, and a reporter. RNP1 binds the genome sequence from the target pathogens, while RNP2 binds the DNA sequence derived from the loop-opening cleavage of the TCC amplifier. **b** Convenient TCC POC workflow. Clinical serum samples are collected (I) and then undergo thermal lysis (II). The lysate is directly placed into a 100 μL TCC mixture (III). The resultant mixture is allowed to react at 37 °C for 30–60 min and isothermal fluorescence is detected to quantify the target gene (IV). **c** Mechanistic scheme of the TCC

reaction. Initially, target binding by RNP1 (I) induces the collateral-cleavage activity of CasΦ (II), leading to concurrent cleavage of the TCC amplifier and the generation of loop-opened products (III). The sequence complementary with gRNA of RNP2 is released from loop-opened product to bind the gRNA (IV) and activate the collateral-cleavage activity of CasΦ in RNP2 (V). The activated CasΦ in RNP1 and RNP2 then repeatedly cleaves the reporter linker between the fluorophore and the quencher, releasing fluorophore for generating fluorescent signal. The process of amplifier cleavage, CasΦ activation and reporter linker cleavage is cycled, enabling exponential isothermal amplification of the fluorescent signal (VI).

the melting temperature ($T_m$) of the toehold region drops drastically, inducing dissociation of the non-target strand and the subsequent formation of a toehold dsDNA capable of activating RNP2 via toehold-mediated strand displacement (TMSD). In the overall TCC reaction (Fig. 1c), RNP1 first specifically targets the pathogen genome with a PAM site, and the activated RNP1 randomly cleaves the linker between the reporter and quencher (hereafter referred to as the reporter linker) and TCC amplifier. The reporter cleavage generates a fluorescence signal for detection, while TCC amplifier cleavage yields a toehold-bearing dsDNA product that activates the *trans*-cleavage activity of RNP2 via TMSD, initiating the next *trans*-cleavage cycle. Consequently, the activated RNP2 exponentially accumulates over time, extensively cleaving the reporter linker and producing an exponential amplification effect. This single-enzyme reaction achieves CCE amplification, enabling one-pot isothermal amplification-free POC diagnosis when combined with our portable diagnostic device (Fig. 1b).

The first step in constructing TCC was to optimize the basic fluorescence detection reaction of the reaction system. We first purified and verified CasΦ protein activity (Supplementary Fig. 1), optimized the reaction system (Supplementary Fig. 2), achieving at least a 24-fold enhancement in the cleavage efficiency of the optimized CasΦ towards the reporter compared to previous reports (Supplementary Fig. 1c)[36]. In the CCE amplification process, the rate of activator dissociation from the amplifier and the *trans*-cleavage efficiency of RNP2 are critical determinants of the reaction kinetics

(Fig. 1c). It was reported that the sequence and base length of the gRNA spacer region crucially influence the *trans*-cleavage efficiency of Cas proteins[37,49,50]. Consequently, to screen for gRNA2 with high *trans*-cleavage rates, we initially designed three gRNAs with distinct spacer sequences (Supplementary Fig. 3a), selecting gRNAa with the highest activation efficiency to assemble RNP2 (Supplementary Fig. 3b, c). Finally, to identify the most suitable activator length for the TCC amplifier, we designed ssDNA activators ranging from 10 to 60 nt (Supplementary Fig. 4a–f). Notably, the 20-mer spacer and 20-nt activator exhibited the highest activation efficiency (Supplementary Fig. 3d–i), corroborating previous findings that long-chain activators can inhibit CasΦ activity[48]. Therefore, we used a 20 nt ssDNA activator as a model for constructing the TCC amplifier. To engineer an amplifier structure with low background, we introduced a blocker sequence fully complementary to the activator for signal suppression, forming a PAM-less dsDNA activator (Supplementary Fig. 5d). The signal suppression ratio was quantified by comparing the *trans*-cleavage efficiencies of the ssDNA activator and the blocked dsDNA activator (Supplementary Fig. 5g). Remarkably, by comparing activators with 5′ and 3′ base end deletion (Supplementary Fig. 5a–f), we discovered that the intact 20 bp dsDNA activator without any truncations exhibited a 23.3-fold signal suppression ratio at concentrations up to 25 nM (Supplementary Fig. 5h), demonstrating optimal signal blockade capability. To validate the signal blockade ability of this PAM-less dsDNA activator, we compared and verified it against

different types of dsDNA activators with PAMs, and the PAM-less dsDNA activator consistently showed the best blockade performance (Supplementary Fig. 6), indicating the feasibility of using this dsDNA activator as a model for constructing the amplifier.

The TCC amplifier, upon being cleaved by CasΦ, ultimately forms dsDNA with a sticky toehold end (Fig. 1c), thereby activating RNP2. Based on this, we first needed to verify whether the activation efficiency of toehold dsDNA was consistent with that of the ssDNA activator. To this end, we designed dsDNA activators with 5' toehold and 3' toehold sticky ends (Supplementary Fig. 7a). Our results demonstrated that the activation efficiency of the 5' toehold dsDNA activator was inferior to that of the 3' toehold dsDNA activator (Supplementary Fig. 7b, c). The activation efficiency of 3' toehold dsDNA with more than 5 nt toehold and ssDNA activators has no difference (Supplementary Fig. 7c) and can achieve maximum activation efficiency. Even blocking or extending the 5' end (Supplementary Fig. 7d) has not any effect. This is consistent with previous results on Cas12a[51,52], indicating that the exchange at 3' toehold promotes dsDNA unwinding and the subsequent R-loop formation[48,53]. Previous research has established that the steric hindrance effect generated by stem-loop structures can effectively suppress Cas protein collateral-cleavage activity[54]. Therefore, we further introduced 2 stem-loop structures for signal blockade, and designed TCC amplifiers with 5'-end and 3'-end toeholds (Supplementary Fig. 8a, b) for CCE reaction (Fig. 1c). The results show that the TCC amplifier with 3' toehold has a faster activation efficiency (Supplementary Fig. 8c), which is consistent with the result that the activation efficiency of 3' toehold dsDNA is higher than 5' toehold (Supplementary Fig. 7). To further verify the feasibility of the CCE reaction, we used 10 nM target-activated RNP1 to *trans*-cleave 500 nM TCC amplifier, and used the loop-opened products of the cleaved TCC amplifier to activate RNP2. The results show that only after the addition of target, the reaction will be significantly activated (Fig. 2a). This indicates that after the TCC amplifier is cleaved, 3' toehold dsDNA activators that can activate RNP2 are formed, demonstrating that our designed amplifier is feasible.

Next, to further optimize the stability and signal-to-noise ratio of the TCC amplifier, we designed amplifiers with loop lengths of 6 nt, 8 nt, 10 nt, and 12 nt to evaluate the CCE reaction (Supplementary Fig. 9), as the stem-loop length is a crucial determinant of the thermodynamic stability of DNA secondary and tertiary structures[55]. Native-PAGE results show the status of 500 nM amplifiers with different stem lengths after 30 min cleavage by RNP1 (Fig. 2b). The 8 nt (lane 3), 10 nt (lane 5) and 12 nt (lane 7) stems are more prone to *trans*-cleavage, releasing more products. The 8 nt loop gives the highest signal-to-noise ratio (Fig. 2c), consistent with the predicted Gibbs free energy by NUPACK (Supplementary Fig. 9). Finally, to assess the efficiency of CasΦ *Trans*-cleaving the TCC amplifier, we determined the cleavage kinetics of 500 nM amplifier by 10 nM activated CasΦ (Fig. 2d). The rate constant $k$ was calculated to be 0.021 min$^{-1}$ (Supplementary Fig. 10a) and $k_{cat}$ to be 0.0175 turnovers s$^{-1}$. With cleavage of both stem-loops required for dissociation, the actual $k_{cat}$ is 0.035 turnovers s$^{-1}$, consistent with using the reporter as the cleavage substrate (Supplementary Figs. 10b, c and 11), indicating the stem-loop can be recognized and cleaved by CasΦ like ssDNA. Theoretically, when only one loop of the amplifier is cleaved, it retains a relatively high $T_m$ value, resulting in three uncleaved states, consistent with the experimental observations (Fig. 2d, right panel, lanes 1–6). Upon cleavage of both stem-loop structures, the product with a toehold is released (lanes 3–8). Concurrently, the CCE reaction verified that the fully cleaved amplifier exhibited the highest activation efficiency (Fig. 2e). Therefore, the 8 nt loop-amplifier exhibits high stability and CCE ability, making it the preferred amplifier in this study for the construction of the TCC method.

## Ultra-sensitive detection of sub-aM level target by TCC

After successful validation of the CCE reaction, to further construct the one-pot TCC reaction, we first optimized the reaction system and components (Supplementary Fig. 12), determining the final TCC reaction system (TCC amplifier 10 nM, CasΦ 30 nM, gRNA1 15 nM, gRNA2 15 nM, reporter 250 nM). We compared and optimized the reactions using two-step and one-step methods to advance the development of the one-pot TCC assay. The two-step method utilized 100 pM target-activated RNP1 to pre-cleave a 10 nM amplifier for 30 min, followed by the addition of 1.5 µL RNP2 (15 nM) for the CCE reaction (Supplementary Fig. 13a). Conversely, the one-step method involved adding all components simultaneously for the reaction (Supplementary Fig. 13b). The results revealed that the one-step method led to faster fluorescence growth. Exponential fitting of the curves showed that the fluorescence growth rate of the one-step method ($y = 10059e^{0.013t}$) was approximately twice that of the two-step method ($y = 5022e^{0.01t}$) (Supplementary Fig. 13c).

To further assess the recognition ability of TCC for dsDNA, we compared the fluorescence detection systems based on TCC and CasΦ. These systems differ only by a TCC amplifier. The system based on CasΦ is constructed similarly to DETECTR[27], with signal amplification relying on the enzymatic efficiency of the protein itself, resulting in a linear accumulation process of reporter cleavage (Supplementary Fig. 13d). It has been reported that the turnover rate of Cas12 subfamily[56,57] activated by low concentration ssDNA or dsDNA activators (1 nM) ranges between 0.022–0.58 turnovers s$^{-1}$, corresponding to 1.32–34.8 turnovers min$^{-1}$. We measured the cleavage efficiency of CasΦ against the TCC amplifier activated by a high concentration DNA activator (10 nM) to be 2.1 turnovers min$^{-1}$ (Supplementary Fig. 10a), indicating a linear accumulation process consistent with the enzymatic efficiency of the Cas12 subfamily. Since each cleaved TCC amplifier releases a product in a cycle to activate another RNP2, we assume that there are $n$ activators activating RNP1, then the number of activated RNPs ($N$) after t min is equal to $N(t) = n(1 + 2.1)^t$. This equation is a typical exponential growth equation and is basically consistent with the fluorescence growth equation we fitted ($y = 10059e^{0.013t}$) (Supplementary Fig. 13c). Therefore, we have characterized the feasibility of TCC theoretically in terms of cleavage efficiency.

*Staphylococcus aureus* (*S. aureus*) ranks among the top five pathogens in global infection and mortality[1]. Its clinical significance calls upon an accurate, rapid, and ultrasensitive method for its diagnosis. To assess the sensitivity and specificity of TCC in detecting *S. aureus* gene fragments, we chose dsDNA fragments of the thermonuclease (*nuc*) gene, recommended in the Clinical and Laboratory Standards Institute (CLSI) guidelines (ISO 21474-1:2020). The *nuc* gene is pivotal for *S. aureus* pathogenicity[58], making it a significant pathogenic factor and marker. Testing TCC's sensitivity and specificity for this gene fragment under optimal conditions, TCC can detect dsDNA fragments down to 0.18 aM (equivalent to 10.8 copies in a 100 µl reaction volume) (Fig. 2f), with a linear range between 0.18 aM to 10 aM (Supplementary Fig. 13e). Compared with CasΦ, TCC displayed at least a 6-order-of-magnitude improvement in sensitivity, demonstrating ultra-sensitivity, which is superior to previously adopted CRISPR-based amplification-free strategies[29,59,60]. Additionally, TCC exhibited a characteristic narrower linear range, suggesting a limitation in the quantitative range due to the nature of the proteins involved[28,29,39]. Moreover, as the TCC amplifier does not require chemical modifications, it may offer a more cost-effective solution. The amplification-free, simplicity, ultra-sensitivity, and cost-effectiveness of the TCC reaction system highlight its potential for clinical applications.

Subsequently, we assessed TCC's single-base discrimination capability by introducing single nucleotide polymorphisms (SNPs) at the PAM site (M1–M3) and proximal to it (M4–M13) (Fig. 2g). Testing TCC

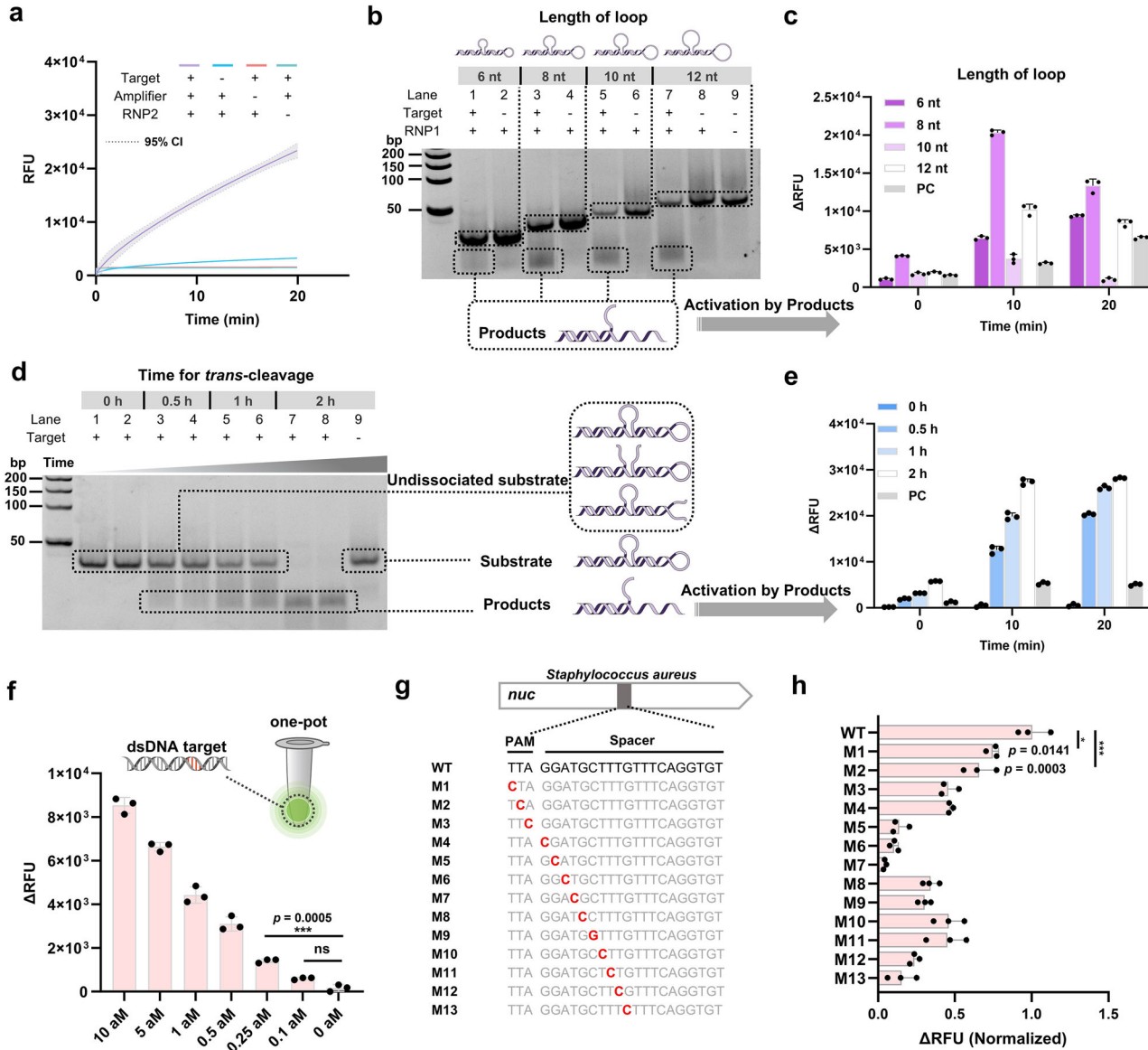

**Fig. 2 | Construction of the Collateral-Cleavage-Enhancing (CCE) reaction in TCC. a** Product activation verification of TTC. The TCC amplifier (500 nM) is *trans*-cleaved by 10 nM target-activated RNP1 (30 nM) for 20 min and its product is then used for activation of RNP2 (30 nM). A robust reaction is observed only when all of the four TCC components (RNP1, target, TCC amplifier and RNP2) are simultaneously present. Curve fitting is performed by taking the average of three technical replicates. Dashed lines represent the 95% confidence interval (95% CI). **b** Screening of TCC amplifier loop lengths. Amplifiers with different loop lengths (6 nt, 8 nt, 10 nt, 12 nt) are subjected to *trans*-cleavage for 30 min and subsequently analyzed via native-PAGE. Product formation is observed only in reactions where the target is included. Reaction conditions remain consistent with those in part a. **c** Product activation verification from b, revealing that the 8 nt loop amplifier

exhibited the optimal signal-to-noise ratio. PC, positive control activated by 100 pM ssDNA activator. **d** Time analysis of *trans*-cleavage for the 8nt loop length amplifier revealed complete substrate cleavage and dissociation of the product at 2 h. **e** Product activation verification from d, products from different cleavage times was performed to validate the cleavage kinetics. **f** Sensitivity of TCC to dsDNA targets. **g** and **h** Single-base recognition properties of TCC for dsDNA targets. Unpaired one-way ANOVA and Turkey's multiple comparisons test were used to statistically analyze each group of independent technical replicates ($n = 3$), where ns indicates no significant difference ($p > 0.05$), and asterisks (*$p \leq 0.05$, **$p \leq 0.01$, ***$p \leq 0.001$, ****$p \leq 0.0001$) indicate significant differences. Data are presented as mean values. Error bars represent mean ± standard deviation SD.

at a dsDNA activator concentration of 2.5 aM revealed its ability to distinguish SNPs. Comparing the endpoint fluorescence values of the SNP mutants (M1–M13) with the wild type (WT) at 2 h indicated TCC's proficiency in SNP discrimination. The WT activator exhibited significantly higher activation efficiency than other SNP activators (Fig. 2h), with better SNP discrimination ability in the seed region (M4-M13) compared to other regions (M1-M3), consistent with most reported Cas12 systems[38,61] and the previously reported wild-type CasΦ-2[36]. Notably, TCC showed improved SNP discrimination relative to CasΦ at each SNP site, possibly due to its exponential amplification feature (Supplementary Fig. 13f). In conclusion, we have successfully

developed TCC with sub-aM level ultra-sensitivity and relatively high specificity, making it a powerful tool for detecting target genome.

## Direct detection of pathogenic genomes by TCC

In order to apply TCC to pathogen diagnosis, cultured *S. aureus* was utilized as a simulated sample and subjected to the fastest high-temperature thermal lysis to obtain its cell lysate for TCC diagnosis[42]. The entire workflow only takes a minimum of 40 min to complete pathogen diagnosis (Fig. 3a). First, thermal lysis of cultured *S. aureus* was performed using different lysis solutions (DEPC $H_2O$, PBS, Tris-HCl) to screen the most suitable lysis solution. Our results show that thermal

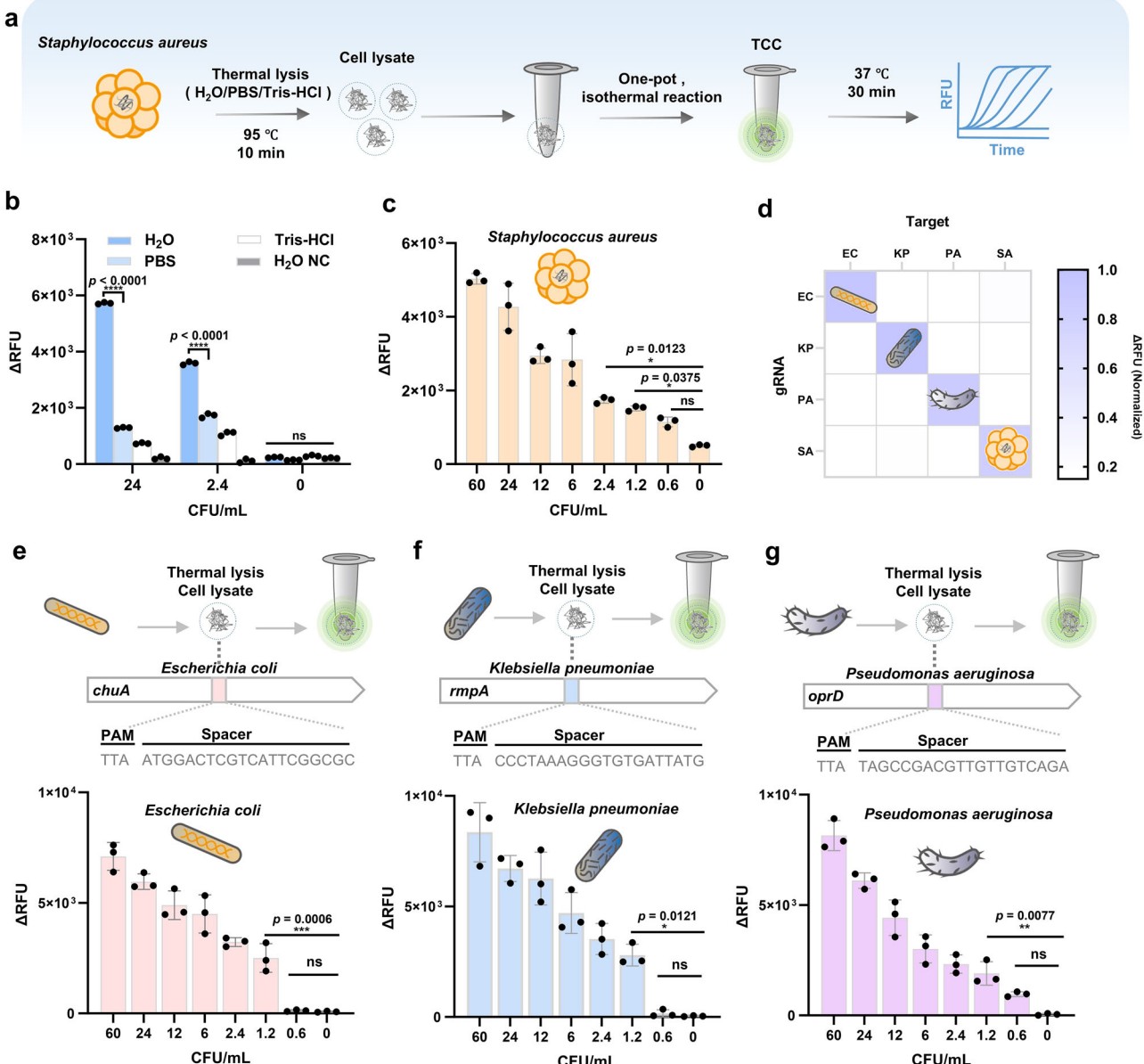

**Fig. 3 | Ultra-sensitivity diagnosis of pathogens by TCC. a** Thermolysis-TCC detection workflow for pathogens (using *S. aureus* as an example). **b** Fluorescence detection showing thermolysis by DEPC $H_2O$ leads to the best signal-to-noise ratio. $H_2O$ NC, negative control by thermolysis of *S. Typhi* in DEPC $H_2O$. **c** Sensitivity of TCC detection for *S. aureus*. **d** Specificity heatmap of TCC detecting pathogens. TCC testing of thermolysis products (24 CFU/mL) from different bacteria using specific gRNAs for *E. coli*, *K. pneumoniae*, *Pseudomonas aeruginosa* (*P. aeruginosa*) and *S. aureus*. EC, *E. coli*. KP, *K. pneumoniae*. PA, *P.*

*aeruginosa*. SA, *S. aureus*. **e** Ultra-sensitivity detection of *E. coli* chuA gene by TCC. **f** Ultra-sensitivity of detecting *K. pneumoniae* rmpA gene by TCC. **g** Ultra-sensitivity of detecting *P. aeruginosa* oprD gene by TCC. Unpaired one-way ANOVA and Turkey's multiple comparisons test were used to statistically analyze each group of independent technical replicates ($n = 3$), where ns indicates no significant difference ($p > 0.05$), and asterisks (*$p \leq 0.05$, **$p \leq 0.01$, ***$p \leq 0.001$, ****$p \leq 0.0001$) indicate significant differences. Data are presented as mean values. Error bars represent mean ± SD.

lysis with DEPC $H_2O$ has the highest signal-to-noise ratio and yields the highest signal under activation of lysate products at different concentrations (Fig. 3b), verifying the feasibility of TCC detecting thermally lysed *S. aureus*. This outcome is expected because high-concentration thermal lysate products contain abundant proteins and nucleic acids. Compared with buffers such as PBS and Tris-HCl, proteins have the lowest solubility in DEPC $H_2O$, thereby reducing interference to TCC from contaminating proteins. Additionally, under high temperature, the DNA superhelical structure unfolds, and most proteins are inactivated. Therefore, such thermal lysate products are more easily retrieved and targeted by Cas proteins[36,62]. Theoretically, as long as the gRNA in CRISPR-CasΦ retrieves and recognizes a genomic target, the TCC reaction will be initiated and cycled to generate an exponentially

amplified fluorescence signal (Fig. 1c). Consequently, TCC can detect *S. aureus* down to 1.2 CFU/mL (Fig. 3c, Supplementary Fig. 14a). In contrast, using the same thermal lysate products as sample input, the sensitivity of qPCR can detect up to $0.6 \times 10^4$ CFU/mL at most (Amplification efficiency = 91.18%) (Supplementary Fig. 14b, c, e), aligning with established commercial qPCR sensitivities ($\sim 0.1 \times 10^4 - 10^5$ copies/mL)[13–15] and MIQE guidelines[6]. Therefore, TCC's sensitivity is superior to qPCR, achieving detection at trace pathogen levels. TCC also exhibits a linear relationship in detecting *S. aureus* from 1.2 to 60 CFU/mL (Supplementary Fig. 14b). We further evaluated the stability and reproducibility of TCC using *Escherichia coli* (*E. coli*) (NC1) and *Salmonella typhi* (*S. typhi*) (NC2) as negative controls among six different measurement batches (with four parallel experiments for each measurement). The intra-assay

coefficients of variation (Intro-assay CV) of TCC range from 4.39% to 8.89%, and the inter-assay CV is 15.85%, demonstrating favorable stability and reproducibility (Supplementary Fig. 14d).

To further evaluate the anti-interference ability and specificity of TCC for pathogen diagnosis, we used specific gRNA for *S. aureus* and simultaneously detected five different pathogens (*Streptococcus pyogenes* (*S. pyogenes*), *Listeria monocytogenes* (*L. monocytogenes*), *E. coli*, and *S. typhi*) with the same workflow. Under the unified lysate concentration of 24 CFU/mL, only *S. aureus* lysate as the sample input exhibited a significant signal (Supplementary Fig. 14f), indicating that TCC detection of different pathogens has high specificity. We also used mixed lysates of different pathogens as sample inputs. Mixture 1 (SA:SP:LM = 1:1:1) and Mixture 2 (SA:EC:ET = 1:1:1) could detect *S. aureus* in 30 min, with no significant difference from the single *S. aureus* input, indicating that TCC has a great anti-interference ability.

Next, to test the broad applicability of TCC for detecting different pathogens, we designed specific gRNAs targeting the outer membrane hemin receptor *chuA* gene of *E. coli*[63], the virulence gene *rmpA* of *Klebsiella pneumoniae* (*K. pneumoniae*)[64], and the outer membrane protein *oprD* gene of *P. aeruginosa*[65]. These pathogens are also among the top five global causes of infections and deaths[1,2]. Using the same workflow as detecting *S. aureus*, we specifically detected four pathogens, and the specificity heatmap showed fluorescence signals only in the presence of lysates from the corresponding pathogens with matched gRNAs (Fig. 3d). Therefore, the specific detection of pathogens can be easily achieved by simple design and replacement of gRNA1 in the TCC system. We consistently realized the ultra-sensitivity detection (1.2 CFU/mL) (Fig. 3e–g) of these three pathogens (Supplementary Fig. 15), which sufficiently demonstrates the broad applicability of TCC for the ultra-sensitivity detection of various pathogens.

## Potential POC diagnosis of serum pathogen infection

Capitalizing on TCC's capability for rapid and ultrasensitive detection of low-concentration pathogens, we endeavored to diagnose potential pathogens in serum samples. The serum samples collected underwent biochemical (VITEK2 system) or MALDI-TOF MS identification after hospital blood culture systems, comprising 24 patients positive for *E. coli* infection, 8 patients positive for *K. pneumoniae* infection, 5 patients positive for *S. aureus* infection, 8 healthy individuals, and 5 patients-positive for other infectious strains. (Supplementary Data 2). Leveraging our experience in sample processing, we optimized the workflow for serum processing, enabling the enrichment of pathogens in serum (Fig. 4a). With a 1 mL serum sample, we ultimately concentrated the thermal-lysed extract into 40 μL, thereby increasing the sample concentration 25-fold compared to the initial concentration. Subsequently, we utilized the same lysis products as input samples and diagnosed *E. coli*-positive and negative patients using TCC and qPCR, respectively. Among the 24 *E. coli*-positive patients, TCC accurately diagnosed 24 positive infections (ΔRFU threshold = 0.511), yielding a sensitivity of 100 % (Fig. 4b, c), whereas qPCR (Amplification efficiency = 103.63%) failed to detect any positive samples (Fig. 4b, c) (Ct = 27.77–31.70, Ct threshold = 26.91). For the 26 *E. coli*-negative patients, TCC diagnosed 23 of 26 negative samples with a specificity of 88.64%. Akin to the diagnosis of positive samples, qPCR failed to make a true negative diagnosis for the 26 negative samples (Fig. 4b, c) (Ct = 26.62–31.88), lacking the sensitivity to diagnose trace pathogens in serum (Supplementary Data 2), and there were 2 false positive diagnoses (Patient ID 33 and 42). The box plot of TCC (Fig. 4d) and qPCR (Fig. 4e) diagnoses concurrently highlighted qPCR's inability to detect low concentrations of pathogens compared to TCC, without forming a linear relationship between TCC–ΔRFU and qPCR-Ct, thereby precluding the diagnosis of positive patients (Fig. 4c). This implied that the concentration of the

extracted serum samples could not reach $0.6 \times 10^4$ CFU/mL and thus could not be directly detected by qPCR (Supplementary Fig. 16a, b). To evaluate the broad application and specificity of TCC in detecting pathogens in serum samples, we similarly performed specific detection of *K. pneumoniae* and *S. aureus* for all samples. The heatmap results revealed that *K. pneumoniae* (Patient IDs 25–32) and *S. aureus* (Patient ID 33–37) were accurately diagnosed as positive by TCC (Fig. 4f), underscoring TCC's high specificity and broad applicability for pathogen detection in serum.

The TCC's quantitative results corroborated that the concentration of *E. coli*-positive infection in serum samples ranged from 1.23 to 156.77 CFU/mL (Fig. 4g, Supplementary Figs. 16c and 17a), and *E. coli*-negative infection in serum samples ranged from 0 to 1.59 CFU/mL (Supplementary Fig. 17b). Moreover, we employed *E. coli*-positive patient IDs 8–10, *K. pneumoniae*-positive patient IDs 26-27, and *S. aureus*-positive patient ID 33, all of which were biochemically and MALDI-TOF MS-confirmed as 100% positive samples, for TCC diagnosis. The corresponding microbiological mass spectrometry data have been deposited in the ProteomeXchange Consortium (PXD057138) via iProX[66,67]. TCC successfully detected all samples within 40 min (Supplementary Fig. 18). Thence, these results substantiated TCC's capability to diagnose and quantify pathogens at ultra-low concentrations in serum samples. To further evaluate TCC's ability to diagnose BSI patients, we compared three methods: TCC, qPCR, and MALDI-TOF MS (combined with VITEK2). Compared to MALDI-TOF MS, the hospital's gold standard for BSI diagnosis, TCC shows a significant advantage in shortening diagnosis time. Compared to qPCR, TCC is advantageous in both analysis time and sensitivity. Receiver operating characteristic (ROC) curve analysis revealed strong diagnostic capability of TCC for *E. coli* BSI patients (AUC = 0.992), surpassing that of qPCR (AUC = 0.5184) (Fig. 4h). In comparison with MALDI-TOF MS, TCC achieved 100% sensitivity, detecting all positive samples, and demonstrated 88.46% specificity, correctly identifying 23 out of 26 *E. coli*-negative patients (Fig. 4i). Collectively, these findings indicate that TCC is a promising diagnostic tool for BSI patients, offering potential advantages in both diagnostic accuracy and efficiency.

For clinical applications, POC diagnostics for ultra-low-concentration pathogens in blood, such as BSI, are highly desirable yet challenging, as they can significantly reduce treatment times. To integrate TCC's favorable diagnostic performance with portable POC diagnostics, we developed a miniaturized fluorescence detection device (termed the TCC Device) as a proof-of-concept for POC applications (Supplementary Fig. 19a). Next, we validated the TCC Device's feasibility using *E. coli*-positive and negative patient samples, alongside a positive control (ΔRFU threshold = 0.644). Fluorescence kinetics revealed a more pronounced increase for positive samples compared to negatives (Supplementary Fig. 19b), indicating that the TCC Device combination could diagnose BSI serum samples. The box plot also showed a significant signal difference between positive and negative samples (Supplementary Fig. 20b), consistent with the BioTek microplate reader results (Fig. 4d). Among the 26 samples, exhibiting 100% concordance with the BioTek microplate reader (Supplementary Fig. 20a, Fig. 4b). Among the 11 *E. coli*-positive samples measured, the fluorescence values obtained by the BioTek microplate reader and the TCC Device showed a high correlation ($R^2 = 0.8162$) (Supplementary Fig. 20c), indicating that the TCC Device also possesses quantification capability for POC diagnosis, successfully quantifying positive *E. coli* in serum samples (Supplementary Fig. 20d). Finally, ROC analysis of the TCC Device, BioTek microplate reader, and qPCR demonstrated that the TCC Device (AUC = 1) and BioTek microplate reader (AUC = 1) yielded consistent diagnostic results for the 26 serum samples, validating TCC's broad applicability (Supplementary Fig. 20e). In contrast, qPCR (AUC = 0.5394) failed to diagnose BSI patients. In summary, by combining the POC diagnostic results of the TCC device, we

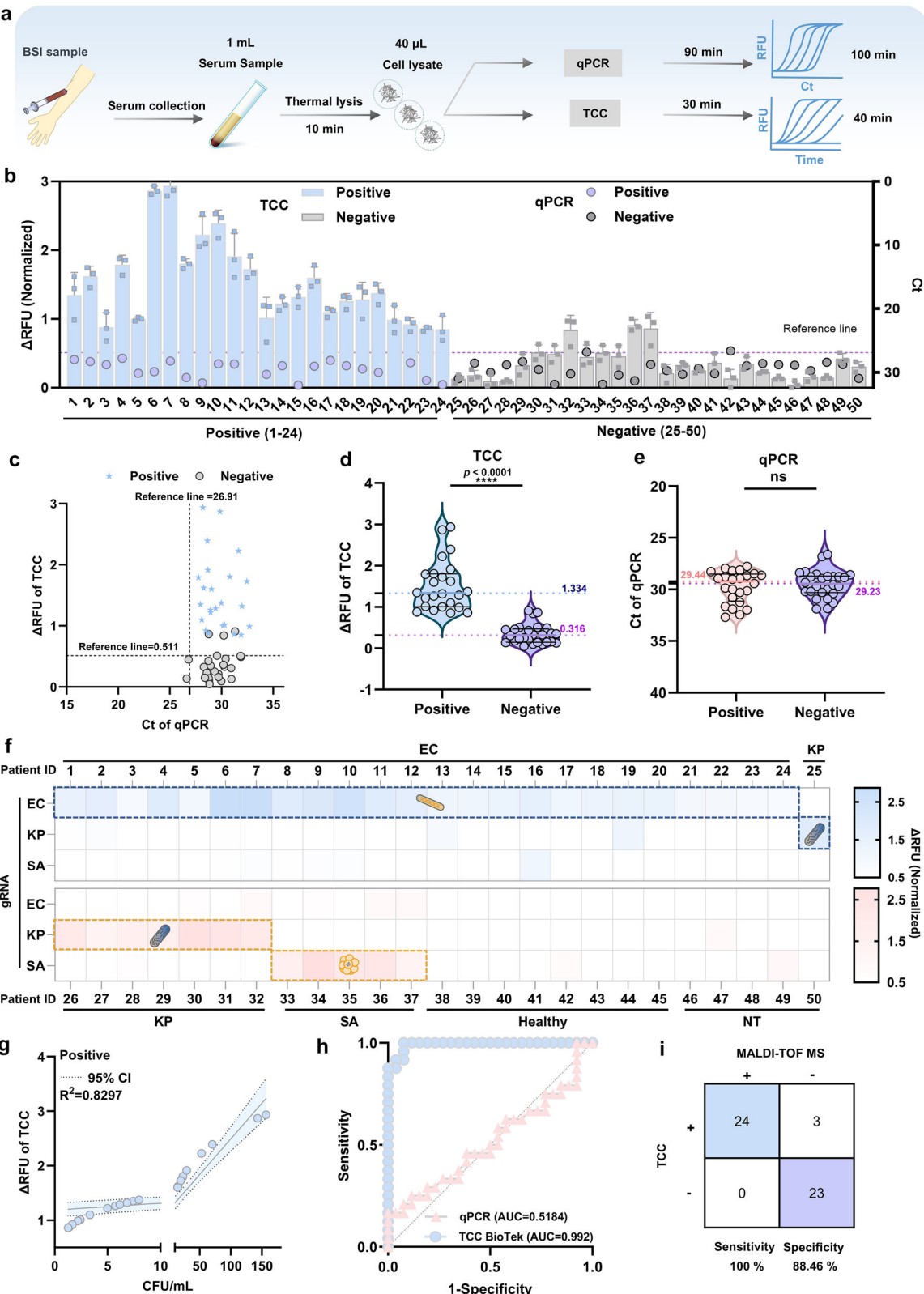

demonstrated its capability and versatility for POC diagnosis, enabling clinical POC diagnosis of BSI with promising prospects for widespread applications.

## Discussion

Rapid and cost-effective POC diagnostics are important for patients with pathogen infections, especially those with BSI. Current blood culture methods can delay treatment for up to 2 days, while detecting pathogens at low concentrations is challenging in hospital settings[12,68]. In this study, we introduce TCC, a simplified diagnostic method for pathogen detection in serum. TCC combines rapid and cost-effective sample processing, enabling rapid, economical, and highly sensitive pathogen detection in serum samples. Compared to gold standard qPCR and blood culture methods, TCC offers superior sensitivity (1.2

**Fig. 4 | Potential of TCC for diagnosis of bloodstream infections (BSI).**
**a** Schematic of diagnostic workflows using TCC and qPCR for BSI patients (using *E. coli* as an example). TCC diagnosis only requires concentrating the serum sample and thermal lysis before direct testing. **b** Clinical diagnosis of BSI patients by TCC and qPCR. Positive *E. coli* infection in 24 patients, represented by Patient IDs 1–24. Negative *E. coli* infection in 26 patients, represented by Patient IDs 25–50. TCC and qPCR reference lines are represented by the same horizontal line. Reference line (TCC), the highest value of negative samples ($\Delta$RFU = 0.511) was used as the reference line. Reference line (qPCR), DEPC $H_2O$ (Ct = 26.91) was used in place of the sample for testing as the reference line. **c** Correlation scatter plot of TCC and qPCR in detecting *E. coli* in serum. **d** box plot comparing TCC diagnosis of positive and negative *E. coli* BSI patients. **e** box plot comparing qPCR diagnosis of positive and negative *E. coli* BSI patients. For d and e, statistical analysis of the diagnostic values for negative and positive patients was performed using an unpaired two-

tailed t-test, where ns = not significant $p > 0.05$, and asterisks (*$p \leq 0.05$, **$p \leq 0.01$, ***$p \leq 0.001$, **$p \leq 0.0001$) indicate significant differences. For box plots in panels d and e, the box represents the interquartile range (IQR), the whiskers extend to 1.5 times the IQR from the box (black line, top: 75%, bottom: 25%), the line within the box represents the median (TCC-positive-blue = 1.334, TCC-negative-purple = 0.316, qPCR-positive-red = 29.44, qPCR-negative-purple = 29.23), and the dots represent individual data points. **f** Heatmap of TCC clinical testing specificity for serum patients. EC, *E. coli*. KP, *K. pneumoniae*. SA, *S. aureus*. NT, Patients positive for other infectious strains. **g** Quantitative assessment of TCC for *E. coli*-positive patients (Patient IDs 1–24). The dashed line represents the 95% CI. **h** ROC performance evaluation of TCC and qPCR for diagnosing *E. coli* in serum samples. **i** TCC comparison of the sensitivity and specific assessment of MALDI-TOF MS. All tests were performed with 3 independent technical replicates ($n = 3$), data are presented as mean values, and error bars represent mean ± SD.

CFU/mL) and faster detection time (as low as 40 min), making it more effective in diagnosing low pathogen concentrations in BSI samples. We have also developed a portable fluorescence detector for on-site validation of TCC, demonstrating consistent results with the BioTek microplate reader using BSI serum samples. This highlights the clinical potential of TCC for POC diagnosis of BSI. TCC's scalability allows for integration into portable platforms, making it a flexible and broadly applicable POC diagnostic tool.

Currently, there is no reliable and rapid method for accurate POC diagnosis of BSI. Particularly for the diagnosis of uncultured BSI blood samples, molecular diagnostic methods based on PCR are almost the only available technology[69,70]. Although it can directly detect common pathogens in blood, the turnaround time exceeds 6 h, and high-resolution real-time PCR instruments are required for melting curve analysis, rendering these PCR-based diagnostic methods challenging for actual POC diagnosis due to their instability[16,17]. The one-pot isothermal reaction of TCC has minimal instrumentation requirements (Fig. 1b), coupled with a rapid serum sample processing procedure (Fig. 4a), allowing for diagnosis and quantification of serum pathogens within a minimum of 40 min (Fig. 4), potentially saving substantial time for diagnosis and treatment of BSI patients. Leveraging the CCE reaction and high specificity of gRNAs for pathogens, TCC can distinguish different pathogens at ultra-low concentrations, with diagnostic sensitivity exceeding PCR-based molecular diagnostic methods. Furthermore, to accurately diagnose BSI patients, hospitals must employ blood culture combined with antimicrobial susceptibility testing and MALDI-TOF MS for characterization, to ensure the authenticity of BSI infections. However, this approach heavily relies on the positive rate of blood culture, and due to the low pathogen load, it is susceptible to contamination during processing, potentially leading to false positives during the culture period exceeding 2 days. TCC eliminates the need for multiple reaction steps, simply requiring a one-pot isothermal reaction with the lysed pathogen sample, minimizing cross-contamination of the sample to the greatest extent. We have verified the practical applicability of TCC for on-site POC diagnosis with the TCC Device (Supplementary Supplementary Fig. 20), potentially alleviating the medical burden of BSI patients in the future.

To our knowledge, TCC has demonstrated remarkable sensitivity among amplification-free biosensing strategies. In recent developments, a variety of amplification-free strategies have emerged, particularly CRIPSR-based technologies. These methods often employ combined several guide RNAs to amplify the sensing signal, leading to a substantial increase in sensitivity levels, reaching up to approximately 100 copies/uL[71]. However, a common challenge encountered in these approaches is the failure in detecting targets with very low concentrations. To overcome this limitation, researchers have integrated an amplifier element, such as an enzyme[29] or a probe[59,72], into CRISPR-based strategies. For example,

the combination of Cas13a and another enzyme (Csm6) represents a notable advancement in this area. In this tandem approach, Csm6 functions as the signal amplifier: the trans-cleaved amplicon product generated by Cas13a activates Csm6, which then cleaves probes to further amplify the signal. However, this strategy, though innovative, only achieved a LOD of approximately 30 copies/uL[29]. To further reduce LOD in orders of magnitude, our proposed TCC method employed a distinct amplifier strategy. We designed a unique amplifier containing two stem-loops for the collateral-cleavage-enhancing CRISPR-Cas$\Phi$ tool. This design makes it possible to continuously activate the collateral-cleavage capability of the Cas$\Phi$ protein through the binding between the Cas/RNP complex and the stem-loop-opened amplifier. The significantly increased activated Cas$\Phi$ protein further collaterally cleaves the linker between the fluorophore and its quencher, releasing the abundant fluorophore to achieve amplified fluorescence signal. The collateral-cleavage enhancement resulted in a remarkable sensitivity, achieving an very low LOD of 0.18 aM (0.11 copies/uL).

Despite the various merits of TCC, further development is required to enable TCC for large-scale pathogen sample diagnosis, as relying solely on a single gRNA for pathogen genome recognition in the TCC assay may be limited. While this study focused on the top five most prevalent pathogens in epidemics[1], diagnosing unknown or uncommon pathogens will require advanced primer or gRNA design, like PCR techniques for large-scale clinical pathogen screening[20,73]. Our method may be extended to the detection of RNA, in particular, the miRNAs. Microfluidic strategies have been suggested to address CRISPR technology limitations and may improve TCC detection[30,39,40], offering high-throughput pathogen diagnostic capabilities. This study provides a reliable method for the diagnosis of clinical BSIs, with potential for developing clinical POC tools. Overcoming the inherent limitations of Cas proteins to achieve wide linear dynamic range, ultra-low LOD, and ultra-high specificity simultaneously is challenging. Combining droplet or microfluidic platforms for signal processing and amplification can help address Cas protein detection deficiencies[4]. Additionally, utilizing general artificial intelligence (AI) to predict interactions between Cas proteins and nucleic acids can enhance specificity and sensitivity by screening for engineered proteins or gRNAs with optimal activation efficiency. In summary, TCC holds significant potential for further development and optimization to meet the needs of biological research, clinical laboratory diagnostics, and POC diagnostics.

## Methods
### Ethical statement
All research was conducted in accordance with relevant guidelines and regulations. This study was approved by the Biomedical Ethics Committee of Haikou People's Hospital (NO. 2024–018). The ethics committee reviewed and approved the submitted project materials (archive control number: SC20240031). The research adhered to the

"Ethical Review Measures for Biomedical Research Involving Humans" (2016) issued by the National Health and Family Planning Commission of China. Additionally, the study was conducted in accordance with the ethical principles outlined in the World Medical Association (WMA) Declaration of Helsinki (2013) and the International Ethical Guidelines for Biomedical Research Involving Human Subjects (2002) jointly developed by the Council for International Organizations of Medical Sciences (CIOMS) and the World Health Organization (WHO). Under a waiver of informed consent, we obtained discarded excess serum samples from patients (A total of 50 human serum samples were randomly collected. Detailed information regarding these samples is provided in Supplementary Data 2). Sex and/or gender were not considered in the study design because de-identified samples were used and no demographic data (including sex/gender, age) were collected. Analysis by sex/gender was not feasible or relevant to our study's objective of evaluating a pathogen detection method. The Medical Testing Center confirmed that the collection of these discarded serum samples did not constitute human subject research. We confirm that, based on the nature of this diagnostic methodology study utilizing publicly available genetic information, approvals from China's Ministry of Science and Technology (MOST) related to the export of genetic information and materials are not required for this publication. All experimental work and data analysis were conducted in accordance with relevant Chinese regulations.

## CasΦ protein expression and purification

The 6×his-CasΦ plasmid (Addgene, #158795) (Supplementary Fig. 1a) was transformed into chemically competent E. coli Trans BL21 (DE3) pLysS cells (transGen Biotech). The cells were cultured in LB medium at 37 °C and 180 rpm until they reached the logarithmic growth phase ($OD_{600} = 0.6$). Subsequently, a final concentration of 0.5 mM IPTG was added, and the cells were incubated at 16 °C and 180 rpm for 16–18 h to induce protein expression. The collected cells were resuspended in lysis buffer (20 mM Tris·HCl, pH 8.0, 500 mM NaCl, 1 mM PMSF) and subjected to sonication on ice. After sonication, the cell lysate was centrifuged at $10,000 \times g$ for 30 min, and the supernatant containing the target protein was collected. This supernatant was then filtered through a 0.22 μm filter (JINTENG) for further processing. The protein supernatant was loaded onto an HT Ni Focurose FF column (HUIYAN-Bio) and subjected to linear elution (0–500 mM imidazole) using elution buffer (20 mM Tris·HCl, pH 8.0, 500 mM NaCl, 500 mM imidazole) in the NGC™ Quest 10 Chromatography System (Bio-rad). Finally, the purified protein was concentrated using an Amicon® Ultra-15 Centrifugal Filter Unit (Merck) and stored at −80 °C in buffer containing 20 mM Tris·HCl (pH 8.0), 500 mM NaCl, and 50% glycerol.

## Design and prediction of nucleic acids

The repeat sequence of the gRNA was referenced to the original sequence[36], and the spacer sequence was designed as the reverse complement of the target. PCR primers were designed online using primer-BLAST from NCBI. All nucleic acid structural predictions were conducted using NUPACK, while $T_m$ values and free energy predictions were determined using DINAMelt, with parameters set at 37 °C, 50 mM Na⁺, and 10 mM Mg²⁺.

## RNA and DNA preparation

All oligos were purchased from Sangon Biotech. All RNA and DNA sequence information is shown in Supplementary Data 1. gRNA, DNA and primers were dissolved in DEPC $H_2O$. For dsDNA activators, target DNA and blocker DNA were annealed in a molar ratio of 1:1 in buffer (10 mM Tris·HCl, pH 8.0, 50 mM NaCl, 5 mM $MgCl_2$). The Annealing conditions were heating in 95 °C water bath for 5 min, then natural cooling to room temperature. For TCC amplifiers, annealing was

performed using the same conditions in annealing buffer (10 mM Tris·HCl, pH 8.0, 150 mM NaCl, 5 mM $MgCl_2$), stored at 4 °C after annealing was completed.

## Trans-cleavage activity assay of CasΦ

In order to form the RNP, CasΦ and gRNA were incubated at a 1:1 molar ratio in 1× reaction buffer (10 mM Tris·HCl, pH 9.0, 50 mM NaCl, 5 mM $MgCl_2$, 100 μg/mL BSA) at 37 °C for 10 min, resulting in a final concentration of 1 μM RNP, which was stored at −80 °C for future use. Fluorescence data were collected at 2-minute intervals. For trans-cleavage activity assays, a 20 μL reaction volume was incubated at 37 °C in a 384-well plate. The reaction mixture included a pre-mixture of 30 nM RNP and 250 nM reporter in 1× reaction buffer (10 mM Tris·HCl, pH 9.0, 50 mM NaCl, 5 mM $MgCl_2$, 100 μg/mL BSA), and the reaction was initiated by adding 25 nM of the target. For DNA/Blocker signal-off detection, a 2.5 nM dsDNA target was used to initiate the reaction. All trans-cleavage reactions were monitored for fluorescence using the BioTek Synergy H1 multimode plate reader (Agilent) with excitation at 492 nm and emission at 520 nm.

## Product activation reaction

Reaction 1 (20 μL volume), with a total volume of 20 μL, comprised 30 nM RNP1, 500 nM TCC amplifier, and 10 nM target, were incubated at 37 °C for 20 min in 1× reaction buffer. Subsequently, the mixture was warmed to 65 °C and aged for 5 min to reach the deactivation of CasΦ. The products (0.4 μL) from reaction 1 were then added to reaction 2 for reactions. Reaction 2 (20 μL volume) included 30 nM RNP2 and 250 nM reporter, and trans-cleavage activity assays were performed using the previously described method.

## Thermal lysis of samples

All cultured pathogenic bacteria were obtained from the China Center of Industrial Culture Collection (CICC) and cultured in LB liquid medium until reaching a concentration of $2.4 \times 10^8$ CFU/mL. Initially, 1 mL of the bacterial suspension was centrifuged at $12,000 \times g$ for 2 min to remove the supernatant. The cells were then resuspended in 100 μL of DEPC $H_2O$, followed by thermal lysis at 95 °C in a water bath for 10 min. After a subsequent centrifugation step at $12,000 \times g$ for 2 min, the supernatant was collected to prepare a lysate sample. For clinical serum samples, 1 mL of serum was initially centrifuged at $800 \times g$ for 5 min to transfer the supernatant. The supernatant was then subjected to a second centrifugation at $12000 \times g$ for 2 min to remove any remaining supernatant. The pellet was resuspended in 1 mL of 1× PBS and washed three times. Finally, the pellet was resuspended in 40 μL of DEPC $H_2O$ for thermal lysis.

## TCC protocol

For the two-step assay, a total reaction volume of 100 μL was used. Initially, 15 nM RNP1 and 10 nM TCC amplifier were mixed with 20 μL of the target in 1× reaction buffer and incubated at 37 °C for 30 min. Subsequently, 15 nM RNP2 and 250 nM reporter were added to the reaction mixture for fluorescence detection at 37 °C. For the one-step assay, a total reaction volume of 100 μL was used. This included the direct mixing of 15 nM RNP1, 15 nM RNP2, 10 nM TCC amplifier, 20 μL of the target, and 250 nM reporter in 1 × reaction buffer. The reaction was then carried out at a constant temperature of 37 °C for fluorescence detection.

## qPCR

According to the Clinical and Laboratory Standards Institute (CLSI) guidelines, the detection criteria and background baseline were determined for qPCR of each pathogen. Fluorescence quantification of thermolysis products was directly performed on LineGene 9600 Plus (Bioer Technology) using TB Green® Premix Ex TaqTM II (TAKARA, RR820A) kits. The reaction volume was 20 μl, containing 1×taq mix,

500 nM upstream primer, 500 nM downstream primer, and 1 μl gradient diluted thermolysis products. DEPC $H_2O$ was used to replace sample input as blank controls to determine the background threshold. The PCR procedure was set as follows: first 95 °C pre-denaturation for 5 min; second 95 °C denaturation for 30 s, 55 °C annealing for 20 s, 72 °C extension for 10 s, 35 cycles; third 72 °C final extension for 10 min. For clinical sample detection, 1 μl thermolysis products were directly used as sample input for qPCR.

## Gel electrophoresis studies

For protein analysis using SDS-polyacrylamide gel electrophoresis (PAGE), 10 μL of the protein sample was loaded onto a 12% SDS-PAGE gel. Electrophoresis was conducted in two steps: initially at 80 V for 20 min, followed by a second step at 180 V for 50 min. Subsequently, the protein was stained using Coomassie Brilliant Blue R-250 (Solarbio). For nucleic acid analysis, 10 μL of the nucleic acid sample was subjected to electrophoresis using a 12% native-PAGE gel at 180 V for 50 min. After electrophoresis, the gel was stained with 4SGelRed (Sangon Biotech) in the dark for 5 min. Both protein and nucleic acid samples were imaged and analyzed using the JS-6000 fully automated gel imaging analysis system (P&Q Science Technology).

## MALDI-TOF MS

Blood cultures were performed on samples from 50 patients who were either positive or negative for *E. coli* or positive or negative for other pathogenic infections. The blood culture was first performed on blood samples for 12–24 h. The resulting colonies from blood culture were used for plate culture again for 12–24 h. Preliminary identification was then performed by the hospitals using morphological, staining and biochemical methods (Antimicrobial susceptibility testing). Meanwhile, single colonies ($n = 1$) were picked and smeared onto MALDI target plates, then 70% formic acid solution was added for natural drying, followed by addition of 1 μl matrix solution (1% HCCA) and drying before detection in the MALDI Biotyper® sirius IVD System (Bruker) (according to the manufacturer's instructions and default settings for microbial identification). Use a blank matrix solution as a negative control, and each sample is identified only once. Samples were analyzed and identified by MBT Compass 4.1 software, and finally MALDI-TOF MS identification was performed by database comparison using IVD MALDI Biotyper 2.3 software (Identification scores were calculated by the software based on pattern matching. A score ≥2.0 was considered as high-confidence species identification, a score between 1.7 and 1.99 was considered as probable genus identification, and a score <1.7 was considered unreliable, according to the manufacturer's recommendations). The MALDI Biotyper Compass software and IVD MALDI Biotyper library are described in detail in the instrument manuals provided by Bruker Daltonics. Further information can be found on the Bruker Daltonics website (https://www.bruker.com).

## TCC device

The TCC Device is a comprehensive system comprising various essential components seamlessly integrated to facilitate its functionality (Supplementary Fig. 19). It includes a light source, microcontroller, temperature control, motor, and display. The excitation light is generated by a 490 nm LED light source, which is filtered through filter 1 to ensure high purity 490 nm excitation light. This light is then reflected by dichroic mirror 1, focused by lens 1, and transmitted through the distal end A of the optical fiber to directly irradiate the sample. As for emission light, the 520 nm fluorescent emission from the sample, along with background light, is collected and coupled into the optical path at the distal end B of the optical fiber. Lens 1 shapes these emissions and background light into collimated beams, which pass through dichroic mirror 1 and a 515–535 nm filter 2 to isolate the 520 nm fluorescence. The converted optical signal is then processed through signal conversion. Lens 1 focuses the filtered light onto the photosensitive surface of the sensor, where it is converted into a photocurrent proportional to the fluorescence intensity. This photocurrent undergoes amplification by a Transimpedance Amplifier (TIA) circuit and is output as a voltage signal. Following buffering, the signal is digitized by the Analog-to-Digital Converter (ADC) and subsequently read out by the microcontroller. The motor driver plays a crucial role in facilitating the movement of the entire optoelectronic detection module. It enables lens 1 to focus onto ports A of optical fibers 1–8, thereby enabling the device to achieve switchable detection channels.

## Statistics and reproducibility

Statistical analyzes were performed in GraphPad Prism 9.4.1, Excel 2016 and image J 1.53. All experiments were conducted with three independent technical replicates ($n = 3$). Details on data presentation and the sample size are included in figure legends. No statistical method was used to predetermine sample size. No data were excluded from the analyzes. The experiments were not randomized. The Investigators were not blinded to allocation during experiments and outcome assessment. For bar charts, Unpaired one-way ANOVA and Turkey's multiple comparisons test were used to statistically analyze each group of independent technical replicates. Exponential-One-phase association was used to fit the average values of three independent technical repeats for fluorescence kinetics curves. For box plots, statistical analysis of the diagnostic values for negative and positive patients was performed using an unpaired two-tailed t-test. For ΔRFU calculation, the background fluorescence (replacing target with DEPC $H_2O$) was subtracted from the endpoint fluorescence at a specific time point. The normalized ΔRFU for each group was the ratio of ΔRFU to the ΔRFU (Positive control), calculated by the formula below.

$$\text{ΔRFU (Normalized)} = \frac{\text{ΔRFU}}{\text{ΔRFU (Positive control)}} \quad (1)$$

$$\text{ΔRFU} = \text{RFU} - \text{RFU (background fluorescence)} \quad (2)$$

The LOD was determined as the RFU without activator (background fluorescence) plus 3 times standard deviation (SD) ($n = 3$). The CFU is determined through LB agar plate culturing of gradient-diluted bacterial suspension. The pathogen LOD by TCC and qPCR was determined based on gradient dilution of thermolysis samples from the cultured pathogens. For clinical samples, we substituted their ΔRFU (Normalized) values into the TCC linear quantitative equation to estimate the CFU/mL.

$$\text{LOD} = \text{RFU(background fluorescence)} + 3\text{SD} \quad (3)$$

For calculation of enzymatic reaction kinetics, the signal was calibrated according to previous reports[56]. Briefly, by subtracting background RFU from RFU at a specific timepoint ($t$) and then dividing by the slope difference between their calibration equations (Supplementary Fig. 11), the cleaved reporter concentration $C(t)$ was obtained, based on which $V_O$ was calculated. By Enzyme kinetics fitting, the Michaelis-Menten equation and $k_{cat}$ were obtained under catalytic site concentration Et = 10 nM.

$$C(t)\,(\text{in nM}) = \frac{\text{RFU}(t) - \text{RFU (background fluorescence)}}{\text{Slope} - \text{Slope(background fluorescence)}} \quad (4)$$

$$V0 = \frac{C(t)\,(\text{in nM})}{t\,(\text{in s})} \quad (5)$$

**Reporting summary**

Further information on research design is available in the Nature Portfolio Reporting Summary linked to this article.

## Data availability

All the original data generated in this study are provided as Source Data. The corresponding microbiological mass spectrometry data have been deposited in the ProteomeXchange Consortium as PXD057138 via iProX. Source data are provided with this paper.

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

## Acknowledgements

This work was financially supported by National Natural Science Foundation of China (No. 42206212 to F.S., No. 42376184 to Y.W., No. 42306225 to Z.Y., No. 42166001 to Y.W., and No. 41866002 to Y.W.), Research Foundation of Hainan University (No. KYQD(ZR)-21128 to F.S.), Hainan Provincial Natural Science Foundation of China (No. 322QN228 to F.S., 322MS032 to B.W.), Hainan Province Science and Technology Special Fund (ZDYF2023GXJS003 to Y.W., ZDYF2022XDNY246 to Y.W., No. ZDYF2025SHFZ061 to Y.W.), Collaborative Innovation Center of Marine Science and Technology, Hainan University (XTCX2022HYB09 to Y.W.), and General Research Fund of Hong Kong (14208723 to C. M.). C. M. would also like to thank Hong Kong Jockey Club Charities Trust for financially supporting JC STEM Lab of Nature-Inspired Precision Medical Engineering. H.C. would like to express gratitude to all the funds and teachers for their support of this work.

## Author contributions

Y.W., H.C. and C.M. conceived and designed the research; Y.W. and C.M. supervised the project; H.C. performed technology development, mechanistic, study, analytical characterization, clinical validation and wrote the manuscript; H.W.H., S.L., L.W., Z.H., Z.F. and T.Y. constructed the POC device; H.H. Y.L. and X.H. collected patient samples and assisted in the clinical studies; H.C., R.L.A., D.Y., H.M, L.H., Y.T., Z.C., Y.M., E.Y.X., Y.W., F.S., B.W. and C.M. analyzed the data and wrote the paper.

## Competing interests

The authors declare no competing interests.

## Additional information

Yi Wan or Chuanbin Mao.

**Peer review information** *Nature Communications* thanks Deming Kong
and the other, anonymous, reviewer(s) for their contribution to the peer
review of this work. A peer review file is available.

