## [Transparent Peer Review File · Nature Communications]

Ultrasensitive Detection of Clinical Pathogens Through a Target-amplification-free Collateral-cleavage-enhancing CRISPR-Cas Φ Tool

Corresponding Author: Professor Chuanbin Mao

Version 1:

Reviewer comments:

Reviewer #1

(Remarks to the Author)

The authors addressed most of the concerns raised by the reviewers, importantly editing and clarifying a few points that led to some misunderstanding upon a first read. I still think the power of their assay is overstated as is the inferiority of qPCR and isothermal methods like RPA/LAMP. For example, "However, their reaction systems are complex and primer design is cumbersome." is not really true, but I guess that's a matter of opinion really. Overall the updates have made the method more clear and more accurately framed in the field. Moving to Nature Communications is better suited for this work in my opinion as it represents improvements to existing approaches. I'm comfortable with it being accepted in the current form and journal.

Reviewer #5

(Remarks to the Author)

This manuscript introduces a CRISPR-Cas Φ -based collateral-cleavage-enhancing (CCE) method for ultrasensitive pathogen detection. The proposed system, TCC, reportedly achieves a detection limit of 0.11 copies/ μ L, significantly improving sensitivity compared to qPCR. It demonstrates potential for rapid, amplification-free point-of-care (POC) diagnosis of bloodstream infections (BSIs).

However, the author's response to the issues raised by reviewer #4 lacked the requisite persuasiveness. Moreover, the novelty of this manuscript is not high enough to be published in Nature Communications. Therefore, I do not recommend the acceptance of this manuscript.

Some of my comments are as follows:

Comments for the response from the author)

Comment 1)

I agree with the assertion that the author's technology shows good detection performance. However, it is hard to accept that it is innovative. The method is too similar to existing techniques to be considered innovative. Furthermore, there is a lack of evidence regarding the mechanism that can produce such remarkable performance.

Comment 2)

The author claims to have developed the world's first non-modified DNA amplifier, but this claim is not appropriate. If the purpose of a DNA amplifier is to amplify signals or nucleic acids, then all primers and probes used in PCR or isothermal nucleic acid amplification are also DNA amplifiers. In this perspective, there are already too many non-modified DNA amplifiers.

Therefore, what is truly important is not that a particular substance is unmodified, but that all components used in the diagnostic process are unmodified. However, the author's technology requires a reporter that has undergone chemical

modification to produce a fluorescent signal.

Comment 3)

It is insufficient to merely alter the target material in order to justify the assertion that the methodology is innovative and worthy of publication in a highly regarded journal such as Nature Communications. Numerous papers have previously been published on the detection of other nucleic acids utilizing the CRISPR-Cas12a or j system.

Comment 4)

The first sentence in the paragraph written to compare with CONAN is not acceptable. The author asserts that the amplifier of TCC is DNA, whereas the amplifier of CONAN is RNA, thereby rendering it more stable. I agree that DNA is a more stable molecule than RNA. However, the component that exhibited a similar function to the amplifier in the TCC is also composed of DNA in CONAN. The author should conduct a comparison of the same role group in each case. In this regard, a more accurate comparison would be to compare the RNA amplifier of CONAN with RNP2 of TCC and to compare the amplifier of TCC with T2 of CONAN. A comparison of equal roles reveals that the parts used as DNA and RNA in TCC are also used as DNA and RNA in CONAN. Therefore, this cannot be considered an advantage of the author's paper. It is not appropriate to set the comparison group in a way that makes the author's paper appear more favorable.

Comment 5)

It is challenging to accept the third assertion presented in the comparative analysis of CONAN technologies. It is hard to accept that a positive feedback circuit of this scale can achieve a sensitivity that exceeds that of PCR and CONAN. In order to outperform CONAN's technology, the mechanism must promote a more positive feedback circuit, or the performance of Cas12j must be superior to that of Cas12a. However, it is generally known that Cas12j has poorer trans-cleavage activity than Cas12a. Furthermore, there is no compelling evidence to suggest that the amplification efficiency of TCC technology is superior to that of CONAN, given the underlying mechanism of TCC's technology.

Comment 6)

The author should have presented a comparative analysis of CALSA technology to substantiate the innovative aspects of this method. However, the three reasons presented in the comparative analysis of CALSA focused on showing the efficacy of TCC technology in diagnostic performance rather than providing evidence of novelty.

Comment 7)

The assertions made in the comparative section with AutoCAR do not provide sufficient evidence to support the novelty of the author's technology. The AutoCAR and the author's TCC technology are too similar to be considered distinct. The only distinction between the two amplifiers is the utilization of a single-stranded hairpin amplifier in lieu of a circular amplifier, with the positive feedback generated by the two amplifiers being identical. As the author indicated, the only advantage of the TCC amplifier over the AutoCAR amplifier is that the synthesis process is less complex. However, the mere substitution of one component for the sole purpose of streamlining the preparation process is deemed insufficient to meet the exceptional differentiation criteria for publication in influential journals. Moreover, given the existence of widely available kits for synthesizing circular-form probes, such as rolling circle amplification (RCA), the difficulty in preparing circular amplifiers is not regarded as a significant disadvantage. This also diminishes the appeal of the authors' TCC technology.

Additional comments to the author)

In addition to the aforementioned comments, I have identified several areas for potential revision to enhance the quality of the paper.

- 1) The complementary DNA in line 103 should be modified to "complementary nucleic acid" or "complementary DNA or RNA." This is because the subjects of the sentence, Cas12 and Cas13, respectively, recognize DNA and RNA.
- 2) From lines 374 to 377, the sentences were duplicated.
- 3) The assertion made in line 418 is unsubstantiated.
- 4) There is no mention of Fig. S3d~l in the manuscript.
- 5) As illustrated in Fig. S4, the free energy of the 60-nucleotide sequence is the lowest among the analyzed sequences utilizing NuPACK. In other words, the 60 nt sequence is the most stable. However, it differs from the author's actual experimental results. In the actual experiment, the results indicated that 20 nt was the most stable. Consequently, the author utilized this length in subsequent experiments. Nevertheless, there is no discussion regarding the discrepancy between the NUPACK results and the actual results. The author should either exclude Fig. S4 from the paper or include a discussion in the text that clearly explains the rationale for conducting this analysis.
- 6) There is no mention of Fig. s5a~c in the manuscript.
- 7) There is no mention of Fig. s5e~f in the manuscript.
- 8) It is unclear why the experiment in Fig. S5 was only conducted in the direction of reducing the base to 20 nt. Based on the results of the experiment in Fig. S3, it can be concluded that the optimal results are between 10 and 30 nt. Accordingly, the author should also examine oligomers that are slightly longer than 20 nucleotides.
- 9) It is confirmed that the trends between the data shown in the supplementary figures do not match. The corresponding data are the data presented in Figure S5e, S8c, and S11c. To be more specific, the concentration of dsDNA activator in Figure S5e is 2.5 nM, and the delta RFU is about 3,000. In contrast, the concentration of dsDNA activator in Figure S8c is 10 pM, which is lower than that in S5e, but the Delta RFU is about 6,000, which is higher than S5e. In addition, the result in Figure S11c shows that the concentration of dsDNA activator is 31.25 nM, which is more than 12 times higher than that in S5e, but the RFU is less than 3,000, which is lower than S5e.
- 10) It is unclear whether the NC in Fig. S13c is a two-step NC or a one-pot NC.
- 11) Please describe the methodology used to obtain the CFU/mL values for each clinical sample.

Reviewer #6

(Remarks to the Author)

In this work, an amplification-free collateral-cleavage-enhancing CRISPR-Cas Φ method, named TCC, was developed. This method converts the linear signal amplification of CRISPR-Cas Φ to exponential amplification, thus significantly increasing the detection sensitivity of CRISPR-based biosensors. The proposed method was successfully used for the ultrasensitive detection of clinical pathogens, showing better performances than conventional PCR. I recommended it to be published in Nature Communications after a revision.

(1) In the proposed TCC method, Cas Φ is used. Can other members in Cas12a family be used in the TCC method? Why is Cas Φ selected?

(2) In some works, CRISPR/Cas12a activation ability of 20 nt ssDNA is differently inhibited by forming dsDNA (Analytical Chemistry, 2023, 95, 15723-15735). In this work, the activation ability is well inhibited by forming a dsDNA containing two loops. Could the authors discuss the reasons? In addition, in almost all experiments, the signal increase is represented by Δ RFU. I hope to see the actual background of the proposed TCC method.

(3) Cleaved RNP2 could activate CRISPR/Cas Φ through toehold-mediated strand displacement (TMSD). Will single-stranded toehold be cleaved by activated RNP1?

(4) In Figure 2a, 10 nM target-activated RNP1 gives no any signal increase, why? 10 nM is a relatively high target concentration.

(5) In Figure 2c, 10 nt stem gives much lower signal increase than 8 nt and 12 nt stems, and its signal increase is even lower than that given by 6 nt stem, which is not consistent with the results in Figure 2b. Could the authors discuss the reasons?

(6) In one-step method, RNP2 will compete with Reporter to be cleaved by activated RNP1. However, the authors demonstrated that the one-step method led to faster fluorescence growth than the two-step method. Could the authors discuss the reasons?

(7) The authors said the proposed TCC method is better than PCR. As we known, the signal amplification factor of PCR exceeds 10^6 . How about the signal amplification factor of TCC method?

Version 2:

Reviewer comments:

Reviewer #5

(Remarks to the Author)

The author has addressed most of the concerns raised by the reviewers. However, the terms "innovative" and "first" still do not seem appropriate to describe this paper. Therefore, we agree to publish this paper in Nature Communications if the author replaces these expressions with more moderate ones. The reason for recommending moderate expressions is as follows.

1. The authors claimed that TCC amplifier is the first DNA amplifier that is not chemically modified. However, there are many technologies that use amplifiers without modification, such as catalytic hairpin assembly, toehold-mediated displacement, DNAzyme, and hybridization chain reaction. Therefore, it is recommended that the authors use a more suitable modifier for TCC technology after reading this paper.

2. The fundamental principle underlying both TCC and CONAN technologies entails the process wherein RNP1/Target1 cuts the amplifier, thereby forming RNP2/Target2. The only difference is that TCC is that the amplifier generates Target2, while CONAN is that the amplifier forms RNP2. After that, both methods amplify the signal through RNP2/Target2.

Therefore, it is not appropriate to characterize this technology as innovative due to the presence of a rudimentary alteration in the reaction sequence. It is recommended that the description be modified to a more moderate expression.

Reviewer #6

(Remarks to the Author)

I am satisfied with the revision. The manuscript can be accepted.

Point-to-point response

Reviewers' Comments:

Reviewer #1 (Comments for the Author):

The authors addressed most of the concerns raised by the reviewers, importantly editing and clarifying a few points that led to some misunderstanding upon a first read. I still think the power of their assay is overstated as is the inferiority of qPCR and isothermal methods like RPA/LAMP. For example, "However, their reaction systems are complex and primer design is cumbersome." is not really true, but I guess that's a matter of opinion really. Overall the updates have made the method more clear and more accurately framed in the field. Moving to Nature Communications is better suited for this work in my opinion as it represents improvements to existing approaches. I'm comfortable with it being accepted in the current form and journal.

Reply:

We are very grateful to Reviewer's feedback.

Reviewer #5 (Comments for the Author):

This manuscript introduces a CRISPR-Cas Φ -based collateral-cleavage-enhancing (CCE) method for ultrasensitive pathogen detection. The proposed system, TCC, reportedly achieves a detection limit of 0.11 copies/ μ L, significantly improving sensitivity compared to qPCR. It demonstrates potential for rapid, amplification-free point-of-care (POC) diagnosis of bloodstream infections (BSIs).

However, the author's response to the issues raised by reviewer #4 lacked the requisite persuasiveness. Moreover, the novelty of this manuscript is not high enough to be published in Nature Communications. Therefore, I do not recommend the acceptance of this manuscript.

Reply:

We are very grateful for your careful review and suggestions, and firmly believe that this work represents an advancement in this field for the reasons listed below:

- (1) For the first time, the LOD of the amplification-free CRISPR-Cas system exceeded 0.18 aM (0.11 copies/ μ L).
- (2) For the first time, a single-stranded oligo DNA amplifier without any chemical modification was developed and applied to the amplification-free CRISPR-Cas system.
- (3) Unlike traditional blood culture with 24 hours pre-cultivation, our CRISPR-Cas Φ -based collateral-cleavage-enhancing method was used to detect bloodstream infections (BSI) directly without pre-enrichment.

1. Comment: I agree with the assertion that the author's technology shows good detection performance. However, it is hard to accept that it is innovative. The method is too similar to existing techniques to be considered innovative. Furthermore, there is a lack of evidence regarding the mechanism that can produce such remarkable performance.

Reply: Thank you for your recognition of the remarkable performance of TCC. As we explained above, our method is indeed innovative. To answer the reviewer's concern on "the lack of evidence regarding the mechanism...", we elucidated the mechanism underlying the high performance of the amplifier from its structural stability. Specifically, the dual stem-loop DNA amplifier (i.e., the TCC amplifier) we used in our method exhibited the highest dynamic stability, as demonstrated by the highest hybridization stability ratio (HSR) shown in **Fig. R1a**. Moreover, as a back-end signal amplifier for biosensing, the double stem-loop amplifier (TCC amplifier) exhibited the highest signal amplification factor (13.5-fold) and the lowest background noise. The signal growth of TCC assay followed the exponential equation $y = 3694.2e^{0.0149t}$ (**Fig. R1b**).

Fig. R1 | **a**, Dynamic stability assessment based on the unique ON-OFF collateral-cleavage activity of Cas Φ . A higher Hybridization stability ratio (HSR) indicates greater stability. **b**, Evaluation of signal amplification capabilities between dual stem-loop and single stem-loop back-end signal amplifiers. T, 10 pM dsDNA activator.

To explain clearly the exponential growth principle of TCC, we also revised the following

sentences: From “Since each cleaved TCC amplifier releases a product to activate another RNP2 in a cycle, the CCE reaction conforms to the mechanism of exponential growth, theoretically verifying the feasibility of one-step, and one-pot method of TCC.”

to

“Since each cleaved TCC amplifier releases a product in a cycle to activate another RNP2, we assume that there are n activators activating RNP1, then the number of activated RNPs (N) after t min is equal to $N(t) = n(1+2.1)^t$. This is a typical exponential growth equation that is basically consistent with the fluorescence growth equation we fitted ($y = 10059e^{0.013t}$) (Fig. S13c). Therefore, we have characterized the feasibility of TCC theoretically in terms of cleavage efficiency.” (Page 6, paragraph 3)

2. Comment: The author claims to have developed the world's first non-modified DNA amplifier, but this claim is not appropriate. If the purpose of a DNA amplifier is to amplify signals or nucleic acids, then all primers and probes used in PCR or isothermal nucleic acid amplification are also DNA amplifiers. In this perspective, there are already too many non-modified DNA amplifiers.

Therefore, what is truly important is not that a particular substance is unmodified, but that all components used in the diagnostic process are unmodified. However, the author's technology requires a reporter that has undergone chemical modification to produce a fluorescent signal.

Reply: Thank you for your comments. According to the definition of biosensors (Ates HC, et al. *Nature Reviews Materials* 2022,7, 887-907)¹, recognition elements at the front-end of biosensing include nucleic acids (such as PCR primers), enzymes, antibodies, and cells. The TCC amplifier is not responsible for recognition but serves solely as a back-end signal amplification and transduction element. Therefore, in the field of amplification-free CRISPR sensing, we emphasize the first development of a DNA amplifier without chemical modification. Namely, unlike the primers and probes in PCR or isothermal nucleic acid amplification, our non-modified DNA amplifier is not a recognition element in the biosensor.

3. Comment: It is insufficient to merely alter the target material in order to justify the assertion that the methodology is innovative and worthy of publication in a highly regarded journal such as Nature Communications. Numerous papers have previously been published on the detection of other nucleic acids utilizing the CRISPR-Cas12a or j system.

Reply: As far as we know, there is only one report on the use of CasΦ (Cas12j) for diagnosis (Kang J-E, et al. *Advanced Science* 2024, n/a, 2402580)². The ability of our amplification-free TCC (Limit of Detection=0.18 aM) is significantly superior to that of the EXP-J amplification method reported in this reference (Limit of Detection=1 fM).

4. Comment: The first sentence in the paragraph written to compare with CONAN is not acceptable. The author asserts that the amplifier of TCC is DNA, whereas the amplifier of

CONAN is RNA, thereby rendering it more stable. I agree that DNA is a more stable molecule than RNA. However, the component that exhibited a similar function to the amplifier in the TCC is also composed of DNA in CONAN. The author should conduct a comparison of the same role group in each case. In this regard, a more accurate comparison would be to compare the RNA amplifier of CONAN with RNP2 of TCC and to compare the amplifier of TCC with T2 of CONAN. A comparison of equal roles reveals that the parts used as DNA and RNA in TCC are also used as DNA and RNA in CONAN. Therefore, this cannot be considered an advantage of the author's paper. It is not appropriate to set the comparison group in a way that makes the author's paper appear more favorable.

Reply: We partially disagree with the reviewer's comparison. The cleaved products of both CONAN and TCC amplifiers are assembled into complexes (T2) for self-cycling. However, the CONAN amplifier consists of RNA and Blocker-DNA (**Table R1**), with the cleaved RNA product, sgRNA, used for cycling (**Fig. R2**). In contrast, the TCC amplifier utilizes the cleaved DNA product, the second target, for cycling.

Notably, the DNA products in the TCC are inherently more stable than the RNA products in the CONAN. Additionally, the RNA product of CONAN requires further folding into gRNA2 to assemble with Cas12a, which introduces additional complexity and potential inefficiencies (**Fig. R2**).

Table. R1 | Components and Sequences of CONAN and TCC Amplifiers.

CONAN amplifier (Science Advances , 2021, 7(5), eabc7802)	Blocker-DNA1: ATCTACACTT/i6FAMdT/TTATT/iBHQ1dT/AGTAGAAATTA
	Blocker-DNA2: TCATAGTTAG/i6FAMdT/TTATT/iBHQ1dT/CGTAACGATC
	Oligo RNA: UAAUUUCUACUAAGUGUAGAUGAUCGUUACGCUA ACUAUGA
TCC amplifier (non-modified) (this work)	Oligo DNA: GGTCTCCTCAAACAAAGCATCCTTTTTTTTGGATGTTTTTTTCT TTGTTTGAGGAGACC

Figure Redacted

Fig. R2 | a, The RNA product of the CONAN RNA amplifier (*Science Advances*, 2021, 7(5), eabc7802). **b**, The DNA product of the TCC DNA amplifier in this work.

5. Comment: It is challenging to accept the third assertion presented in the comparative analysis of CONAN technologies. It is hard to accept that a positive feedback circuit of this scale can achieve a sensitivity that exceeds that of PCR and CONAN. In order to outperform CONAN's technology, the mechanism must promote a more positive feedback circuit, or the performance of Cas12j must be superior to that of Cas12a. However, it is generally known that Cas12j has poorer trans-cleavage activity than Cas12a. Furthermore, there is no compelling evidence to suggest that the amplification efficiency of TCC technology is superior to that of CONAN, given the underlying mechanism of TCC's technology.

Reply: As mentioned above, the RNA product of CONAN is gRNA2, which requires further folding and assembly to form RNP2 before it can bind the target. This additional step introduces lower reaction efficiency, necessitating more reaction time. Indeed, our preliminary studies have shown that the activation efficiency of unfolded-gRNA+Cas+target is significantly lower compared to that of pre-formed RNP+target (**Fig. R3**). Furthermore, the inherently unstable RNA product is prone to degradation, potentially generating non-functional gRNA. In contrast, the DNA product of TCC can directly activate RNP2, resulting in higher activation efficiency and greater stability.

Fig. R3 | Investigation of the impact of CasΦ:gRNA:target incubation sequence on activation efficiency. RNP+T: Target directly activates pre-incubated RNP. Cas+gRNA+T: No pre-formed RNP; Cas, gRNA, and target are activated in one step. gRNA:T + Cas: No pre-formed RNP; gRNA is first incubated with the target, followed by the addition of Cas for activation.

The reviewer stated that the cleavage activity of CasΦ (Cas12j) is inferior to that of Cas12a, which is untrue. Different enzymes have different cleavage efficiencies in different buffers. For the two buffers we tested (**Fig. R4**), the k_{cat} of CasΦ can range from 2.1 to 16.5 min^{-1} , which is consistent with the reported k_{cat} values of the Cas12 subfamily (1.32-34.8 min^{-1}) (Huyke DA, et al. *Analytical Chemistry*, 2022, 94, 9826 - 9834)³ and (Ramachandran A, et al. *Analytical Chemistry*, 2021, 93, 7456 - 7464)⁴.

Fig. R4 | The enzymatic reaction efficiency of CasΦ in Tris-HCl and Tris-Acetate.

In addition, the detection ability of TCC is significantly better than that of CONAN (Shi K, et al. *Science Advances* 2021, 7, eabc7802.)⁵ (**Table. R2**).

Regarding the reaction mechanism, we have clearly illustrated and verified the exponential growth principle of TCC in the main text:

“Exponential fitting of the curves showed that the fluorescence growth rate of the one-step method ($y=10059e^{0.013t}$) was approximately twice that of the two-step method ($y=5022e^{0.01t}$) (**Fig. S13c**).”

Table. R2 | Comparison of the Detection Abilities of TCC and CONAN

Name	TCC	CONAN
Exponential equation of signal	$y=10059e^{0.013t}$	$y=696e^{0.012t}-693.9$
Sample processing	Thermal-lysis	Kit extraction
Amplifier stability	$T_m=72.2^\circ\text{C}$, $\Delta G=-17.08$ kcal/mol	$T_m=41.7^\circ\text{C}$, $\Delta G=-10.1$ kcal/mol
Amplifier Modification	No	Yes
Annealing of oligos	single-oligo	triple-oligo
length of oligos	60 nt	41/28/27 nt
Activator	Produced DNA from amplifier	Produced RNA as gRNA
Products activation principle	TMSD ⁶	Targeted activation
LOD	0.18 aM	5 aM
sample to answer	40 min	3h

6. Comment: The author should have presented a comparative analysis of CALSA technology to substantiate the innovative aspects of this method. However, the three reasons presented in the comparative analysis of CALSA focused on showing the efficacy of TCC technology in diagnostic performance rather than providing evidence of novelty.

Reply: Apparently, the detection ability and stability of TCC are comprehensively ahead of CALSA (Sun K, *et al. Nucleic Acids Research*, 2024, 52(7), e39-e39)⁷, as we detailed in **Table R3**.

Table. R3 | Comparison of the Detection Abilities of TCC and CALSA

Name	TCC	CALSA
Exponential equation of signal	$y=10059e^{0.013t}$	No
Sample processing	Thermal-lysis	Kit extraction
Amplifier stability	$T_m=72.2^\circ\text{C}$, $\Delta G=-17.08$ kcal/mol	$T_m=33.2^\circ\text{C}$, $\Delta G=-9.8$ kcal/mol (by DINAMelt)
Amplifier Modification	No	Yes
Annealing of oligos	single-oligo	single-oligo
length of oligos	60 nt	11 nt
Products activation principle	TMSD	Targeted activation
LOD	0.18 aM	25 fM
sample to answer	40 min	60 min

7. Comment: The assertions made in the comparative section with AutoCAR do not provide sufficient evidence to support the novelty of the author's technology. The AutoCAR and the

author's TCC technology are too similar to be considered distinct. The only distinction between the two amplifiers is the utilization of a single-stranded hairpin amplifier in lieu of a circular amplifier, with the positive feedback generated by the two amplifiers being identical. As the author indicated, the only advantage of the TCC amplifier over the AutoCAR amplifier is that the synthesis process is less complex. However, the mere substitution of one component for the sole purpose of streamlining the preparation process is deemed insufficient to meet the exceptional differentiation criteria for publication in influential journals. Moreover, given the existence of widely available kits for synthesizing circular-form probes, such as rolling circle amplification (RCA), the difficulty in preparing circular amplifiers is not regarded as a significant disadvantage. This also diminishes the appeal of the authors' TCC technology.

Reply: In practice, the circular amplifier requires four steps: annealing of three oligos, T4 ligation, exonuclease treatment, and annealing (Fig. R5). It is difficult to balance the reaction efficiency of each of these steps. Moreover, the detection ability and stability of AutoCAR is also lower than that of TCC, as we summarized in Table R4.

Figure Redacted

Fig. R5| The circular amplifier requires four steps: annealing, T4 ligation, exonuclease treatment, and annealing. (Method 1, Deng F, et al. *Nature Communications*, 2024, 15(1), 1818)⁸.

Table. R4| Comparison of the Detection Abilities of TCC and AutoCAR

Name	TCC	AutoCAR
Exponential equation of signal	$y=10059e^{0.013t}$	$y=3.64172e^{0.0167t}$
Sample processing	Thermal-lysis	No
Amplifier stability	$T_m=72.2^{\circ}\text{C}$, $\Delta G=-17.08$ kcal/mol	$T_m=53.5^{\circ}\text{C}$, $\Delta G=-18.2$ kcal/mol (by DINAMelt)
Amplifier Modification	No	Yes
Annealing of oligos	single-oligo	triple-oligo
length of oligos	60 nt	20/18 nt and linker
Products activation principle	TMSD	Targeted activation
LOD	0.18 aM	1 aM
sample to answer	40 min	60 min

8. Comment: The complementary DNA in line 103 should be modified to "complementary nucleic acid" or "complementary DNA or RNA." This is because the subjects of the sentence, Cas12 and Cas13, respectively, recognize DNA and RNA.

Reply: We sincerely appreciate your valuable correction. We have revised "DNA" to "DNA or RNA". (Page3, paragraph3)

9. Comment: From lines 374 to 377, the sentences were duplicated.

Reply: Thank you for your careful review. We have already deleted the repetitive sentences. (Page10, paragraph2)

10. Comment: The assertion made in line 418 is unsubstantiated.

Reply: We are very grateful for your meticulous review. We have already incorporated the references, and the corrected sentence is given as follows:

"...like PCR techniques for large-scale clinical pathogen screening^{20, 73}" (Page 11, paragraph 2)

11. Comment: There is no mention of Fig. S3d-I in the manuscript.

Reply: We have corrected this part. The newly revised sentence is as follows:

"Notably, the 20-mer spacer and 20-nt activator exhibited the highest activation efficiency (Fig. 3d-i)" (Page 4, paragraph 3)

12. Comment: As illustrated in Fig. S4, the free energy of the 60-nucleotide sequence is the lowest among the analyzed sequences utilizing NuPACK. In other words, the 60 nt sequence is the most stable. However, it differs from the author's actual experimental results. In the actual experiment, the results indicated that 20 nt was the most stable. Consequently, the author utilized this length in subsequent experiments. Nevertheless, there is no discussion regarding the discrepancy between the NUPACK results and the actual results. The author should either exclude Fig. S4 from the paper or include a discussion in the text that clearly explains the rationale for conducting this analysis.

Reply: Thank you for your correction. We have re-discussed this part. The newly revised sentence is as follows:

"corroborating previous findings that long-chain activators can inhibit CasΦ activity" (Page4, paragraph3)

13. Comment: There is no mention of Fig. s5a~c in the manuscript. There is no mention of Fig. s5e~f in the manuscript.

Reply: We appreciate your review and have corrected this part. The newly revised sentence is as follows:

"The signal suppression ratio was quantified by comparing the *trans*-cleavage efficiencies of the ssDNA activator and the blocked dsDNA activator (Fig. S5g). Remarkably, by comparing activators with 5' and 3' base end deletion (Fig. S5a-f), we discovered that the intact 20 bp dsDNA activator without any truncations exhibited a 23.3-fold signal suppression ratio at concentrations up to 25 nM (Fig. S5h)." (Page5, paragraph1)

14. Comment: It is unclear why the experiment in Fig. S5 was only conducted in the direction of reducing the base to 20 nt. Based on the results of the experiment in Fig. S3, it can be concluded that the optimal results are between 10 and 30 nt. Accordingly, the author should also examine oligomers that are slightly longer than 20 nucleotides.

Reply: Apparently, the results in Fig. R6 (Fig. S3g-i) indicate that the 20-nt activator represents the highest activation efficiency.

Fig. R6 (Fig. S3g-i) | g, Schematic diagram of ssDNA activator length optimization screening. **h** and **i**, Measurement of *trans*-cleavage efficiencies of activated Cas Φ by ssDNA activators with different lengths, showing 20nt ssDNA activator has the best efficiency.

15. Comment: It is confirmed that the trends between the data shown in the supplementary figures do not match. The corresponding data are the data presented in Figure S5e, S8c, and S11c. To be more specific, the concentration of dsDNA activator in Figure S5e is 2.5 nM, and the delta RFU is about 3,000. In contrast, the concentration of dsDNA activator in Figure S8C is 10 pM, which is lower than that in S5e, but the Delta RFU is about 6,000, which is higher than S5e. In addition, the result in Figure S11c shows that the concentration of dsDNA activator is 31.25 nM, which is more than 12 times higher than that in S5e, but the RFU is less than 3,000, which is lower than S5e.

Reply: This observation is due to the difference in the fluorescence time points, activators, reaction batches and experiment types for the three different figures, as explained in Table R5. Additionally, in Fig. S11c, the concentration of 31.25 nM refers to the FQ-reporter, not the dsDNA activator.

Table. R5 | Comparison of different experiments.

Name	Fig. S5e	Fig. S8c	Fig. S11c
Experiment type	Collateral-cleavage	Collateral-cleavage	Catalytic efficiency
Time of Δ RFU	20 min	120 min	4 min
Activator	no PAM-dsDNA 2.5 nM	ssDNA 10 pM	PAM-dsDNA 10 nM
FQ concentration	250 nM	250 nM	31.25 nM
Δ RFU	~3000	~6000	~1000

16. Comment: It is unclear whether the NC in Fig. S13c is a two-step NC or a one-pot NC.

Reply: We have corrected this part. "NC" stands for "one-pot NC".

17. Comment: Please describe the methodology used to obtain the CFU/mL values for each clinical sample.

Reply: We initially determined the linear equation for TCC quantification using simulated *Escherichia coli* samples: $\Delta\text{RFU (Normalized)} = 10^{(0.255 \cdot \log(\text{CFU/mL}) - 0.09217)}$. Subsequently, the CFU/mL of clinical samples can be estimated by substituting the measured $\Delta\text{RFU (Normalized)}$ values of the clinical samples into this equation. The estimation method is consistent with those reported (Wang Y, et al. *Nature Communications* 2024, 15, 3279; Deng F, et al. *Nature Communications*, 2024, 15, 1818)^{8,9}.

Reviewer #6 (Remarks to the Author):

In this work, an amplification-free collateral-cleavage-enhancing CRISPR-Cas Φ method, named TCC, was developed. This method converts the linear signal amplification of CRISPR-Cas Φ to exponential amplification, thus significantly increasing the detection sensitivity of CRISPR-based biosensors. The proposed method was successfully used for the ultrasensitive detection of clinical pathogens, showing better performances than conventional PCR. I recommended it to be published in *Nature Communications* after a revision.

1. Comment: In the proposed TCC method, Cas Φ is used. Can other members in Cas12a family be used in the TCC method? Why is Cas Φ selected?

Reply: Other Cas12a family members may require customized amplifiers tailored to their ON-OFF collateral-cleavage activity. The TCC amplifier is specifically designed for Cas Φ . Based on our findings (Fig. R1, Fig. S8), Cas Φ is unable to recognize the PAM sequence (5'-TTN-3') within the 3' end loop structure, which distinguishes it from Cas12a and Cas9^{10, 11, 12} (Bagheri N, et al. *Angewandte Chemie International Edition* 2024, n/a, e202319677) (Zuo T, et al. *Nucleic Acids Research* 2024, gkae804) (O'Connell MR, et al. *Nature* 2014, 516, 263-266).

Additionally, Cas Φ exhibits collateral-cleavage efficiency comparable to that of Cas12a (Fig. R3) and offers the advantage of simpler purification, making it the preferred choice for this study.

2. Comment: In some works, CRISPR/Cas12a activation ability of 20 nt ssDNA is differently inhibited by forming dsDNA (*Analytical Chemistry*, 2023, 95, 15723-15735). In this work, the activation ability is well inhibited by forming a dsDNA containing two loops. Could the authors discuss the reasons? In addition, in almost all experiments, the signal increase is represented by ΔRFU . I hope to see the actual background of the proposed TCC method.

Reply: We appreciate your review and constructive suggestions. We have supplemented the discussion on the signal-blocking capability of the TCC amplifier in the main text:

"Previous research has established that the steric hindrance effect generated by stem-loop structures can effectively suppress Cas protein collateral-cleavage activity⁵⁴."

Furthermore, we have demonstrated the actual background levels (RFU) of TCC in **Fig. R1b** (dual-T), **Fig. R7** (Fig. S15), etc.

Fig. R7 (Fig. S15) | Sensitivity of TCC in diagnosing different pathogenic bacteria. 0: DEPC H₂O was used as a blank control in place of sample input.

3. Comment: Cleaved RNP2 could activate CRISPR/CasΦ through toehold-mediated strand displacement (TMSD). Will single-stranded toehold be cleaved by activated RNP1?

Reply: This phenomenon was observed in our other study on single stem-loop amplifiers (**Fig. R6**). Briefly, the single stem-loop structure requires only one loop cleavage to release the toehold, whereas the dual stem-loop structure necessitates two cleavages. As a result, the probability of toehold cleavage in the double stem-loop amplifier is significantly lower than that in the single stem-loop amplifier.

Fig. R6 | Native-PAGE characterization of the cleavage kinetics of the single stem-loop amplifier. The product activation kinetics reveal that the 30-min product exhibits the fastest activation efficiency. Beyond 30-60 minutes, the activation efficiency linearly decreases due to the cleavage of the toehold.

4. Comment: In Figure 2a, 10 nM target-activated RNP1 gives no any signal increase, why? 10 nM is a relatively high target concentration.

Reply: This is because we employed a two-step method for feasibility characterization. As

detailed in the "Product activation reaction" section of the Methods: " the mixture (RNP1) was warmed to 65°C and aged for 5 min to reach the deactivation of CasΦ."

5. Comment: In Figure 2c, 10 nt stem gives much lower signal increase than 8 nt and 12 nt stems, and its signal increase is even lower than that given by 6 nt stem, which is not insistent with the results in Figure 2b. Could the authors discuss the reasons?

Reply: We sincerely appreciate your thorough review and valuable suggestions. As indicated by the product grayscale values in Fig. 2b, the yield of the 10 nt loop product is lower than that of the 8 nt and 12 nt loops, resulting in the lowest activation efficiency. Considering that loops with a length exceeding 10 nt have a higher false positive (RFU), the ΔRFU (10 nt) = RFU (+target) – RFU (-target) is the lowest.

6. Comment: In one-step method, RNP2 will compete with Reporter to be cleaved by activated RNP1. However, the authors demonstrated that the one-step method led to faster fluorescence growth than the two-step method. Could the authors discuss the reasons?

Reply: In the two-step method, the products cleaved by RNP1 cannot further cyclically activate RNP2. In contrast, although the one-step method involves competitive cleavage, the activated RNP2 continues to initiate the next round of cleavage cycles. Therefore, under the same 30-minute treatment duration, the one-step method achieves faster activation efficiency.

7. Comment: The authors said the proposed TCC method is better than PCR. As we known, the signal amplification factor of PCR exceeds 10^6 . How about the signal amplification factor of TCC method?

Reply: We performed theoretical calculations using the data from Fig. 2f as an example. In a 100 μL reaction volume, TCC can detect 0.25 aM of dsDNA. Thus, the number of copies (TCC target) = $100 \mu\text{L} \times 0.25 \text{ amol/L} \times 6.02 \times 10^{23} \text{ mol}^{-1} = 15$. Substituting ΔRFU (0.25 aM)=1409 into the RFU-FQ linear equation: $\text{RFU} = 139 \times \text{FQ} + 530.2$ (Fig. S11a, b), we calculated the concentration of TCC cleaved-FQ to be 6.32 nM. Therefore, the number of copies (TCC cleaved-FQ)= $100 \mu\text{L} \times 6.32 \text{ nmol/L} \times 6.02 \times 10^{23} \text{ mol}^{-1} = 3.8 \times 10^{10}$. Consequently, the amplification factor of TCC, fold (TCC)= $\text{copies (TCC cleaved-FQ)} / \text{copies (TCC target)} = 3.8 \times 10^{10} / 15 = 2.5 \times 10^9$, which surpasses conventional PCR ($\sim 10^6$).

Additionally, given that the cleavage efficiency of CasΦ is 2.1 min^{-1} , the number of cleaved-FQ copies via 15 copies activated-CasΦ within 120 minutes is calculated as $\text{copies (CasΦ cleaved-FQ)} = 15 \text{ copies} \times 120 \text{ min} \times 2.1 \text{ min}^{-1} = 3780$. Finally, the relative amplification factor of TCC compared to CasΦ, fold (TCC/CasΦ)= $\text{copies (TCC cleaved-FQ)} / \text{copies (CasΦ cleaved-FQ)} = 3.8 \times 10^{10} / 3780 = 1 \times 10^7$, which is highly consistent with our results in Fig. 13e (sensitivity $> 10^6$ -fold).

References cited in response letter:

1. Ates HC, *et al.* End-to-end design of wearable sensors. *Nature Reviews Materials* **7**, 887-907 (2022).
2. Kang J-E, *et al.* Unveiling Cas12j Trans-Cleavage Activity for CRISPR Diagnostics: Application to miRNA Detection in Lung Cancer Diagnosis. *Advanced Science* **n/a**, 2402580 (2024).
3. Huyke DA, Ramachandran A, Bashkirov VI, Kotseroglou EK, Kotseroglou T, Santiago JG. Enzyme Kinetics and Detector Sensitivity Determine Limits of Detection of Amplification-Free CRISPR-Cas12 and CRISPR-Cas13 Diagnostics. *Analytical Chemistry* **94**, 9826-9834 (2022).
4. Ramachandran A, Santiago JG. CRISPR Enzyme Kinetics for Molecular Diagnostics. *Analytical Chemistry* **93**, 7456-7464 (2021).
5. Shi K, *et al.* A CRISPR-Cas autocatalysis-driven feedback amplification network for supersensitive DNA diagnostics. *Science Advances* **7**, eabc7802 (2021).
6. Yurke B, Turberfield AJ, Mills AP, Simmel FC, Neumann JL. A DNA-fuelled molecular machine made of DNA. *Nature* **406**, 605-608 (2000).
7. Sun K, *et al.* An autocatalytic CRISPR-Cas amplification effect propelled by the LNA-modified split activators for DNA sensing. *Nucleic Acids Research* **52**, e39-e39 (2024).
8. Deng F, *et al.* Topological barrier to Cas12a activation by circular DNA nanostructures facilitates autocatalysis and transforms DNA/RNA sensing. *Nature Communications* **15**, 1818 (2024).
9. Wang Y, *et al.* Ultrasensitive single-step CRISPR detection of monkeypox virus in minutes with a vest-pocket diagnostic device. *Nature Communications* **15**, 3279 (2024).
10. O'Connell MR, Oakes BL, Sternberg SH, East-Seletsky A, Kaplan M, Doudna JA. Programmable RNA recognition and cleavage by CRISPR/Cas9. *Nature* **516**, 263-266 (2014).
11. Bagheri N, Chamorro A, Idili A, Porchetta A. PAM-Engineered Toehold Switches as Input-Responsive Activators of CRISPR-Cas12a for Sensing Applications. *Angewandte Chemie International Edition* **n/a**, e202319677 (2024).
12. Zuo T, *et al.* FRAME: flap endonuclease 1-engineered PAM module for precise and sensitive modulation of CRISPR/Cas12a trans-cleavage activity. *Nucleic Acids Research*, gkae804 (2024).

REVIEWERS' COMMENTS

Reviewer #5 (Remarks to the Author):

The author has addressed most of the concerns raised by the reviewers. However, the terms “innovative” and “first” still do not seem appropriate to describe this paper. Therefore, we agree to publish this paper in Nature Communications if the author replaces these expressions with more moderate ones. The reason for recommending moderate expressions is as follows.

Reply:

Thank you very much for the reviewers' insightful comments and suggestions. We have moderated the description of our research's novelty by adopting a more cautious tone, as detailed in our point-by-point responses below.

1. Comment: The authors claimed that TCC amplifier is the first DNA amplifier that is not chemically modified. However, there are many technologies that use amplifiers without modification, such as catalytic hairpin assembly, toehold-mediated displacement, DNAzyme, and hybridization chain reaction. Therefore, it is recommended that the authors use a more suitable modifier for TCC technology after reading this paper.

Reply:

We thank the reviewers for their comment. As described in the abstract and main text, our DNA amplifier is a "*dual-stem-loop DNA signal amplifier*" that is not chemically modified.

2. Comment: The fundamental principle underlying both TCC and CONAN technologies entails the process wherein RNP1/Target1 cuts the amplifier, thereby forming RNP2/Target2. The only difference is that TCC is that the amplifier generates Target2, while CONAN is that the amplifier forms RNP2. After that, both methods amplify the signal through RNP2/Target2. Therefore, it is not appropriate to characterize this technology as innovative due to the presence of a rudimentary alteration in the reaction sequence. It is recommended that the description be modified to a more moderate expression.

Reply:

We appreciate the reviewers' insightful feedback. We concur that both methodologies rely on RNP1/Target1-mediated amplifier cleavage, resulting in RNP2/Target2-driven signal amplification.

Adhering to the reviewers' feedback, we have adopted "progressive" to replace "innovative," emphasizing the TCC amplifier as a further **improvement** on CRISPR-based biosensing amplifiers, as elaborated in the manuscript: "*by designing and optimizing a DNA (with dual stem-loop blocking domains) that serves as an amplifier.*" (page 3, paragraph 3)